# High-resolution cryo-EM of the human CDK-activating kinase for structure-based drug design

Victoria I. Cushing [1], Adrian F. Koh [2], Junjie Feng[1], Kaste Jurgaityte [3], Alexander Bondke[4], Sebastian H. B. Kroll[4], Marion Barbazanges [4,6], Bodo Scheiper[4], Ash K. Bahl[5], Anthony G. M. Barrett [4], Simak Ali [3] ✉, Abhay Kotecha [2] ✉ & Basil J. Greber [1] ✉

Rational design of next-generation therapeutics can be facilitated by high-resolution structures of drug targets bound to small-molecule inhibitors. However, application of structure-based methods to macromolecules refractory to crystallization has been hampered by the often-limiting resolution and throughput of cryogenic electron microscopy (cryo-EM). Here, we use high-resolution cryo-EM to determine structures of the CDK-activating kinase, a master regulator of cell growth and division, in its free and nucleotide-bound states and in complex with 15 inhibitors at up to 1.8 Å resolution. Our structures provide detailed insight into inhibitor interactions and networks of water molecules in the active site of cyclin-dependent kinase 7 and provide insights into the mechanisms contributing to inhibitor selectivity, thereby providing the basis for rational design of next-generation therapeutics. These results establish a methodological framework for the use of high-resolution cryo-EM in structure-based drug design.

The human CDK-activating kinase (CAK) is a heterotrimeric protein complex formed by CDK7, cyclin H, and MAT1 and acts as a master regulator of cell growth and division[1,2]. It regulates transcription initiation and the cell cycle by phosphorylating RNA polymerase II and cyclin-dependent kinases, respectively. Due to this central role in cellular physiology, the CAK has been identified as a promising target for cancer therapeutics[3] and it is a possible target for antivirals[4]. Several groups have discovered specific inhibitors of the CAK[4–9], including high-affinity inhibitors that occupy the active site to compete with adenosine-nucleotide binding, and covalent inhibitors that modify a cysteine residue in the vicinity of the active site. We have developed a CDK7 inhibitor series and advanced one of these compounds, ICEC0942 (CT7001, samuraciclib)[5], into clinical trials for treatment of

advanced-stage solid cancers (clinicaltrials.gov NCT04802759, NCT03363893). To date, five other CDK7 inhibitors have entered clinical trials: SY-1365 (NCT03134638), SY-5609 (NCT04247126, NCT04929223), LY-3405105 (NCT03770494), Q901 (NCT05394103) and XL-102 (NCT04726332). One of the major challenges in the development of CDK-targeting compounds is specificity[10] because 20 different enzymes in human cells are classified as CDKs and often exhibit high sequence identity near their catalytic sites[11,12]. Furthermore, CDKs are characterized by large conformational changes upon association with their activatory cyclins and phosphorylation of their regulatory T-loops, which can change their compound-binding properties[13]. To enable the discovery and rational design of next-generation therapeutics with increased potency and reduced off-

[1]The Institute of Cancer Research, Chester Beatty Laboratories, 237 Fulham Road, London SW3 6JB, UK. [2]Materials and Structural Analysis Division, Thermo Fisher Scientific, Achtseweg Noord 5, 5651 Eindhoven, The Netherlands. [3]Division of Cancer, Department of Surgery & Cancer, Imperial College London, Hammersmith Hospital Campus, London, UK. [4]Department of Chemistry, Imperial College London, London, UK. [5]Carrick Therapeutics, Nova UCD, Bellfield Innovation Park, Dublin 4, Ireland. [6]Present address: Institut Parisien de Chimie Moléculaire, Sorbonne Université, CNRS, 4 Place Jussieu, 75252 Paris Cedex 05, France. ✉e-mail: simak.ali@imperial.ac.uk; abhay.kotecha@thermofisher.com; basil.greber@icr.ac.uk

target effects, structural data permitting the application of structure-based drug design approaches are instrumental.

Working towards the goal of making the human CAK accessible to structure-based drug design, we previously determined its cryo-EM structure bound to nucleotide analogs and the inhibitors ICEC0942 and THZ1[14,15]. Our ICEC0942-bound structure revealed interesting conformational differences between the inhibitor bound to its clinical target, CAK, and bound to CDK2, an off-target complex with the potential to cause side effects in patients, highlighting the potential of using structural data to guide the development of next-generation inhibitors in this system[14,16]. Our published cryo-EM structures used the human proteins and have reached higher resolution than X-ray crystal structures of a homologous complex from a thermophilic fungus[17]. Furthermore, one advantage of cryo-EM is the ability to capture a series of different states from a single sample, which can facilitate the structural analysis of dynamic systems. This may be particularly beneficial for CDKs due to their conformation-dependent compound-binding properties[13]. These considerations indicate that cryo-EM is the method of choice in this system if throughput and resolution reach the levels required for drug discovery applications.

In this work, we structurally characterize complexes of CAK bound to a range of both commercially available molecules and the series of compounds developed and characterized alongside ICEC0942, aiming to uncover the structural basis of CDK7 inhibitor selectivity to pave the way towards next-generation therapeutics. The nature of this endeavor requires high resolution for this small, asymmetric complex as well as high throughput, creating a technical challenge and situating this effort at the edge of what is currently feasible using cryo-EM[18]. To address this challenge, we use rapid screening on a 200 kV cryo-transmission electron microscope (cryo-TEM) to obtain initial structures and identify promising specimens, and we progress suitable specimens to high-end data collection on an energy-filtered 300 kV cryo-TEM with a cold field emission gun (cold-FEG). We obtain 18 structures of the 85 kDa CDK-cyclin-module of the human CAK in its nucleotide-bound, free, and inhibitor-bound states at up to 1.8 Å resolution, with the inhibitors comprising pyrazolopyrimidine, pyrazolotriazine, and phenylaminopyrimidine-class compounds. In addition to achieving high resolution from large datasets, we establish a routine ~4 Å and ~3 Å-resolution structure determination workflow for ligand-bound complexes using the 200 kV setup combined with on-the-fly data processing from 1 h and 4 h of data collection. This work provides a workflow for the application of cryo-EM to structure-based drug discovery, expands our understanding of how structurally diverse inhibitors interact with the active site of CDK7, and thereby provides a basis for the design of next-generation cancer therapeutics.

## Results

### High-resolution structures of the human CAK

Structure-based drug design greatly benefits from high-resolution data that provide molecular models with atomic accuracy and information on networks of water molecules present in the active sites of target complexes. Knowledge of ordered water positions is crucial for understanding how inhibitors bind[19] and can guide efforts in inhibitor optimization[20,21]. Our previous structure of nucleotide-bound human CAK reached 2.8 Å resolution[15], limiting its ability to resolve many ordered waters, and an apo-structure of the fully assembled human CAK is lacking altogether. We therefore set out to determine the high-resolution structures of apo- and nucleotide-bound human CAK using the latest generation Krios cryo-TEM[22]. Furthermore, as a proof of concept for high-resolution structure determination of inhibitor-bound CAK, we re-determined the structure of CAK modified by the covalent inhibitor THZ1 (Supplementary Fig. 1a), which has been shown to disrupt super-enhancer transcription in human cancer cell lines[8], and we determined the structure of CAK in complex with the high-affinity inhibitor LDC4297 (Supplementary Fig. 1b), a

pyrazolotriazine-class compound that exhibits anti-viral activity[4], a property that is of particular interest given current global health challenges.

These experiments resulted in cryo-EM reconstructions of the ligand-bound human CAK at 1.9–2.1 Å resolution (Fig. 1a-c, Supplementary Fig. 1c–e, Supplementary Table 1). The maps show protein side chain and main chain features consistent with the reported resolution (Fig. 1d, e), allow direct identification of post-translational modifications, such as N-terminal protein acetylation (Fig. 1f), and reveal the locations of many water molecules (Fig. 1d–f). At high resolution, density for the regulatory CDK7 T-loop (residues 155-182 between the conserved DFG and APE motifs) becomes fragmented, indicating structural heterogeneity.

The structure of CAK bound to the nucleotide analog ATPγS at 1.9 Å resolution allows the accurate placement of the bound nucleotide and a water-coordinated $Mg^{2+}$ ion in the active site (Supplementary Fig. 1f). Compared to the X-ray crystal structure of the homologous complex from *Chaetomium thermophilum*, the orientation of the adenosine base in the bound nucleotides is different, adopting a *syn*-conformation in the fungal complex (Supplementary Fig. 1g)[17]. This conformation is rare but has been observed previously in the structure of homoserine kinase[23]. It is currently unclear if the *syn*-conformation in the fungal complex is the preferred conformation in catalytically activated complexes, which could point to subtle differences in the nucleotide binding pockets between the fungal and human enzyme that impact their small molecule-binding properties, or if the *syn*-conformation in the *C. thermophilum* structure is favored only in the context of the protein conformation compatible with the crystal lattice. The structure of apo-CAK shows great similarity to the nucleotide- and inhibitor-bound structures, even for most of the active site pocket, except for the β-sheet in the N-terminal kinase lobe, which is substantially more flexible in the apo state than in all structures that have an occupied active site (Supplementary Fig. 1h–j). Flexibility limits the overall resolution of this reconstruction to 2.3 Å (Supplementary Fig. 1k) and is greatest at the outermost two β-strands, connected by the G-rich loop (residues 19-24, sequence GEGQFA), which are barely visible in the density. Ligand-dependent conformational changes of the N-terminal kinase lobe have been observed previously in cryo-EM structures of the human CAK[15], X-ray crystal structures of the homologous *C. thermophilum* complex[17], and X-ray crystal structures of a range of other kinases, including CDK2[24] and cAMP-dependent protein kinase, where parts of this domain are displaced by almost 10 Å[25]. However, unlike X-ray crystal structures, where mobile domains can be trapped in specific conformations due to interactions within the crystal lattice and thereby manifest as defined conformations, cryo-EM structures can capture the full range of conformations accessible to molecular complexes in solution. This can explain the fragmented density observed in our cryo-EM map of apo-CAK.

The structure of THZ1-bound CAK at 1.9 Å resolution (Fig. 1a–c, Supplementary Fig. 1c) agrees with our previous interpretation at lower resolution[15]. Notably, even in our high-resolution map, the extended arm that harbors the reactive acrylamide group that covalently modifies CDK7 residue C312 is resolved far less well than the indole and chlor-opyrimidine groups of the inhibitor that are embedded in the CDK7 active site pocket (Supplementary Fig. 2a, b). This is probably linked to continuous flexibility of this part of the inhibitor, a notion that is supported by the observation that the equivalent cysteine-reactive functional groups of two copies of the related inhibitor THZ531 are positioned 12 Å apart in the X-ray crystal structure of this compound bound to CDK12[26]. We quantified these differences in local atom resolvability using Q-scores (Supplementary Fig. 2c)[27] because the local map quality in the presence of inhibitor heterogeneity does not necessarily closely correlate with the overall resolution of the reconstruction.

In contrast to THZ1, which acts via a covalent mechanism, LDC4297 is a highly selective reversible inhibitor of CDK7[4]. The map

quality of our structure of CAK in complex with LDC4297 at 2.1 Å resolution (Supplementary Fig. 1d) was improved by locally refining only the approximately 40 kDa CDK7 density (Supplementary Fig. 2d). The structure reveals that the substituted phenyl ring assumes a similar conformation to the equivalent group in ICEC0942, while the ether-linked piperidine group was refined into a boat-like conformation, though structural flexibility and sampling of other conformations cannot be excluded (Supplementary Fig. 2e).

Knowledge of the positions of bound water molecules in macromolecular complexes is essential to understand inhibitor binding and to explore possibilities for inhibitor optimization[19–21]. Therefore, the visualization of water molecules is an important feature of our high-resolution cryo-EM maps. A systematic validation of our cryo-EM-based assignments by comparing them to the locations of water densities identified with X-ray crystal structures is challenging due to the lack of structures of CDK7 or cyclin H at resolutions of 2 Å or better. However, we find that many of the water locations identified in our cryo-EM maps correspond to those found in CDK7 and cyclin H at lower resolution or in related CDKs at high resolution (Supplementary Fig. 2f, g), supporting the reliability of our assignments. Nevertheless, when compared to X-ray crystallographic structures at similar resolutions, we observe that our cryo-EM structures appear to resolve a smaller total number of water molecules (approx. 160 as compared to nearly 400 in PDB ID 6ATH at 1.8 Å resolution[28]). This has been observed previously for cryo-EM reconstructions at atomic resolution, but the reasons for this discrepancy are currently not known[29].

## Design of a cryo-EM workflow for high-throughput structure determination of inhibitor-bound CAK complexes

ICEC0942 is an orally bioavailable high-affinity inhibitor of CDK7 that has entered clinical trials for cancer treatment[5]. ICEC0942 was developed as a part of a series of pyrazolo[1,5-a]pyrimidine compounds.

In-vitro kinase inhibition data show that inhibitor selectivity spans almost 4 orders of magnitude, ranging from approximately 100× selectivity for CDK7 over CDK2 (ICEC0829) to 100× selectivity for CDK2 over CDK7 (BS-194) (Fig. 2a, Supplementary Table 2)[5,16,30–33], with low selectivity conferring a risk of inducing undesirable off-target effects in patients. Given the importance of understanding selectivity, and considering our previous observation of conformational differences of ICEC0942 bound to CDK7 and CDK2[14], which suggests that structural analysis may aid in rationalizing inhibitor selectivity and provide insights that may be applied to next-generation inhibitors, we sought to undertake the structural analysis of 12 pyrazolopyrimidine-type CDK inhibitors (Fig. 2a) to explore the molecular mechanisms of CDK7 selectivity.

To enable high-resolution and high-throughput cryo-EM data collection suitable for structure determination of small-molecule-bound CAK complexes, we devised and implemented a three-stage cryo-EM workflow comprising rapid initial grid and sample screening, intermediate-resolution structure determination, and finally high-resolution structure determination of the best grids (Fig. 2b–e). We applied this workflow to 12 compounds (Fig. 2a, Supplementary Tables 1, 3, 4), but the approach can be scaled to even larger numbers of small-molecule ligands in the future.

To aid in the design of this workflow, we first benchmarked the performance of a 300 kV Krios G4 cryo-TEM equipped with a Falcon 4i detector, a Selectris X energy filter, and a cold-FEG against a 200 kV Glacios 2 cryo-TEM equipped with an X-FEG, a Falcon 4i detector, and a Selectris X energy filter. We found that for the CDK-activating kinase, the 300 kV-cryo-TEM system provided an approximately 0.3 Å resolution advantage (Fig. 3a–e), with resolutions of 2.0 and 2.3 Å achieved for data collected from the same grid, notably in a shorter time and from approximately 50% fewer electron micrograph movies for the 300 kV system compared to the 200 kV system (see

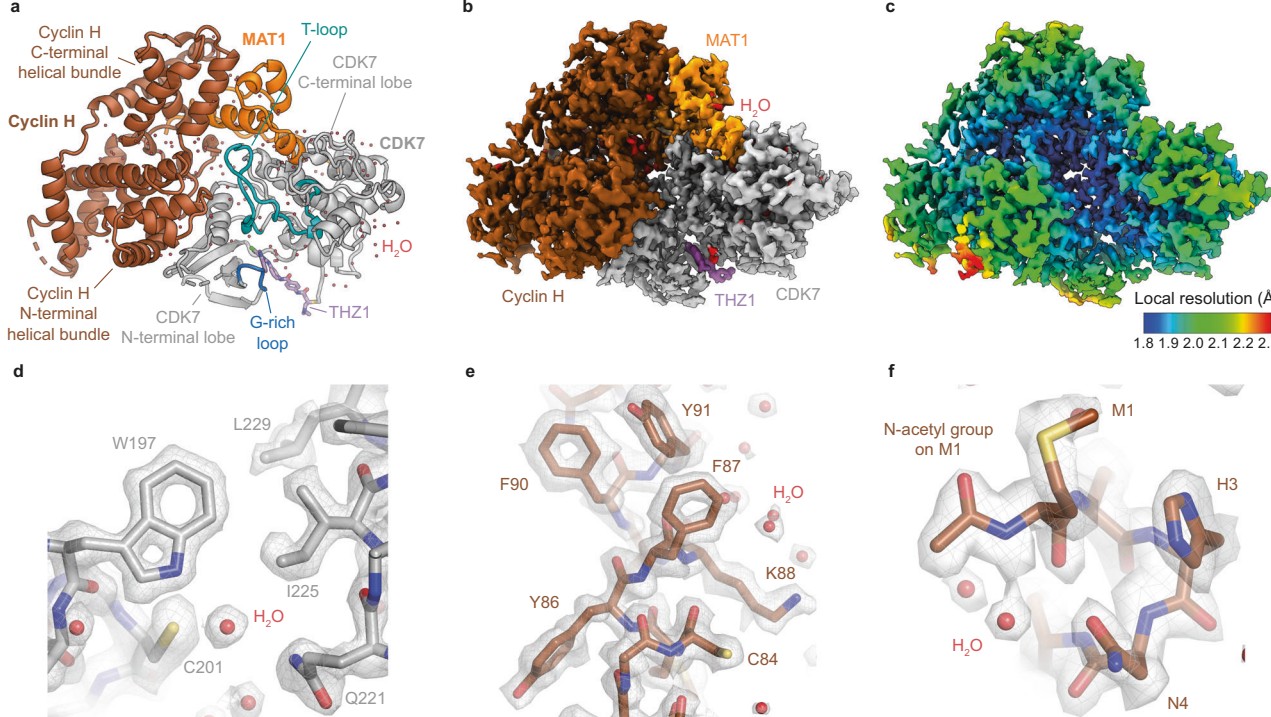

**Fig. 1 | High-resolution structures of the human CDK-activating kinase.**
**a** Atomic model of the CAK-THZ1 adduct with distinctly colored subunits (cyclin H brown, MAT1 orange, CDK7 gray, THZ1 violet). The domains of CDK7 and cyclin H are indicated and the T-loop and G-rich loop of CDK7 are shown in cyan and blue, respectively. Water molecules are shown in red. **b** Cryo-EM map of the CAK-THZ1 adduct. **c** Local resolution plot for the cryo-EM map shown in (**b**). **d**, **e** Examples of high-quality density at 1.9 Å resolution (CAK-THZ1 map). **f** Direct identification of post-translational N-terminal acetyl modification on cyclin H in the cryo-EM map (CAK-ATPγS map).

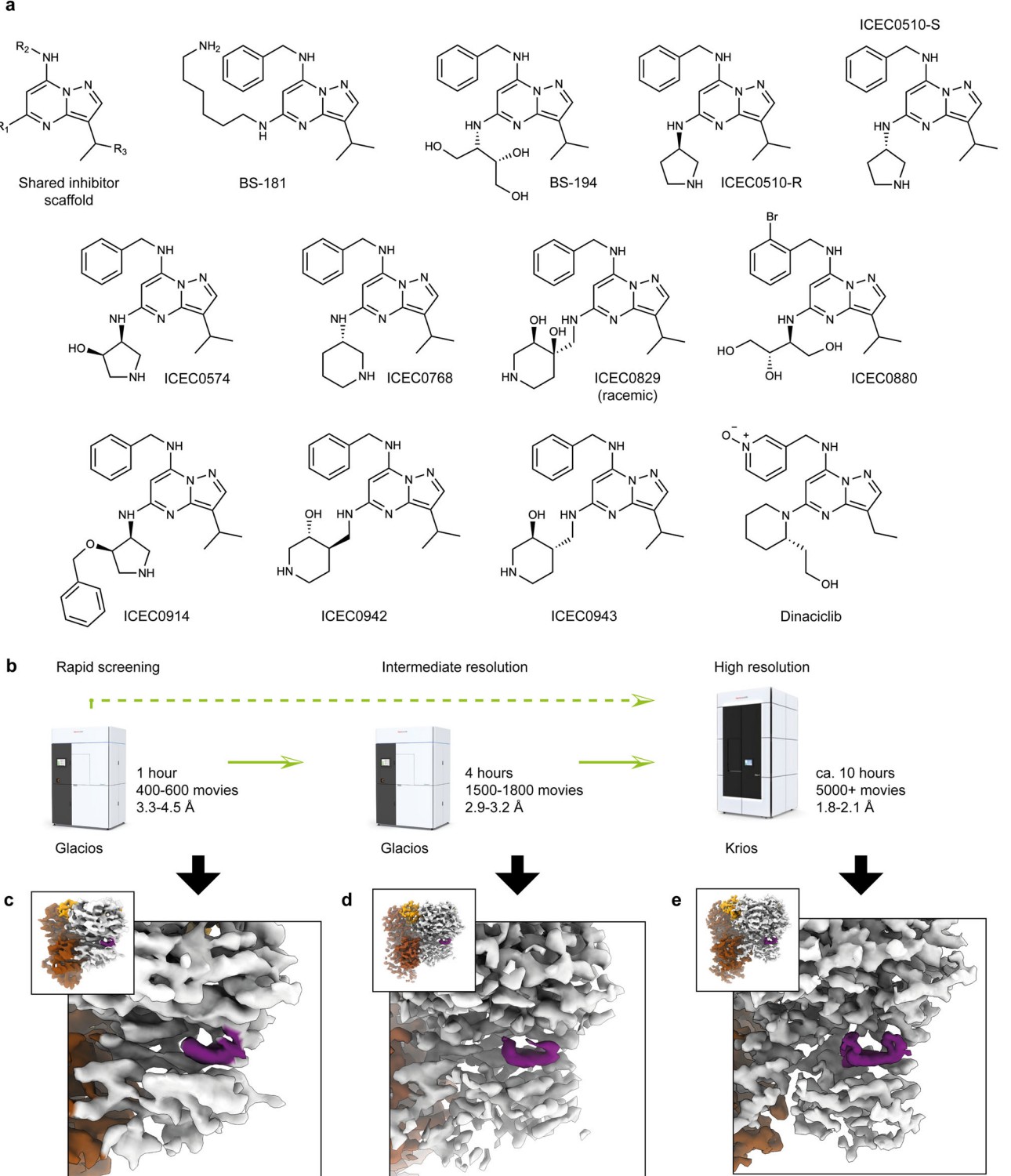

**Fig. 2 | Cryo-EM workflow for inhibitor series structure determination.**
**a** Chemical structures of pyrazolopyrimidine-type inhibitors used for cryo-EM structure determination. The chemical core shared by all inhibitors is shown in the first position, along with definition of the $R_1$-$R_3$ substituent positions that will be used in the text. **b** Schematic of tripartite cryo-EM workflow. The possibility of using a two-stage workflow is indicated by a dashed green arrow. **c** Example result from 1-h Glacios screening with visible ligand density (purple). **d** Example result from 4-h Glacios screening with visible ligand density (purple). **e** Example high-resolution cryo-EM map from Krios G4 collection with visible ligand density (purple).

Supplementary Note 1). We also investigated the contribution of the energy filter to high-resolution structure determination of our 85 kDa target complex. We found a 0.3 Å resolution loss in the absence of energy filtration under the conditions tested (Fig. 3a, f, g, Supplementary Note 2). Given these results, we decided to use the 200 kV Glacios 2 setup for rapid sample screening and intermediate-resolution structure determination, and to leverage the 300 kV Krios G4 setup for collection of high-quality datasets on selected specimens.

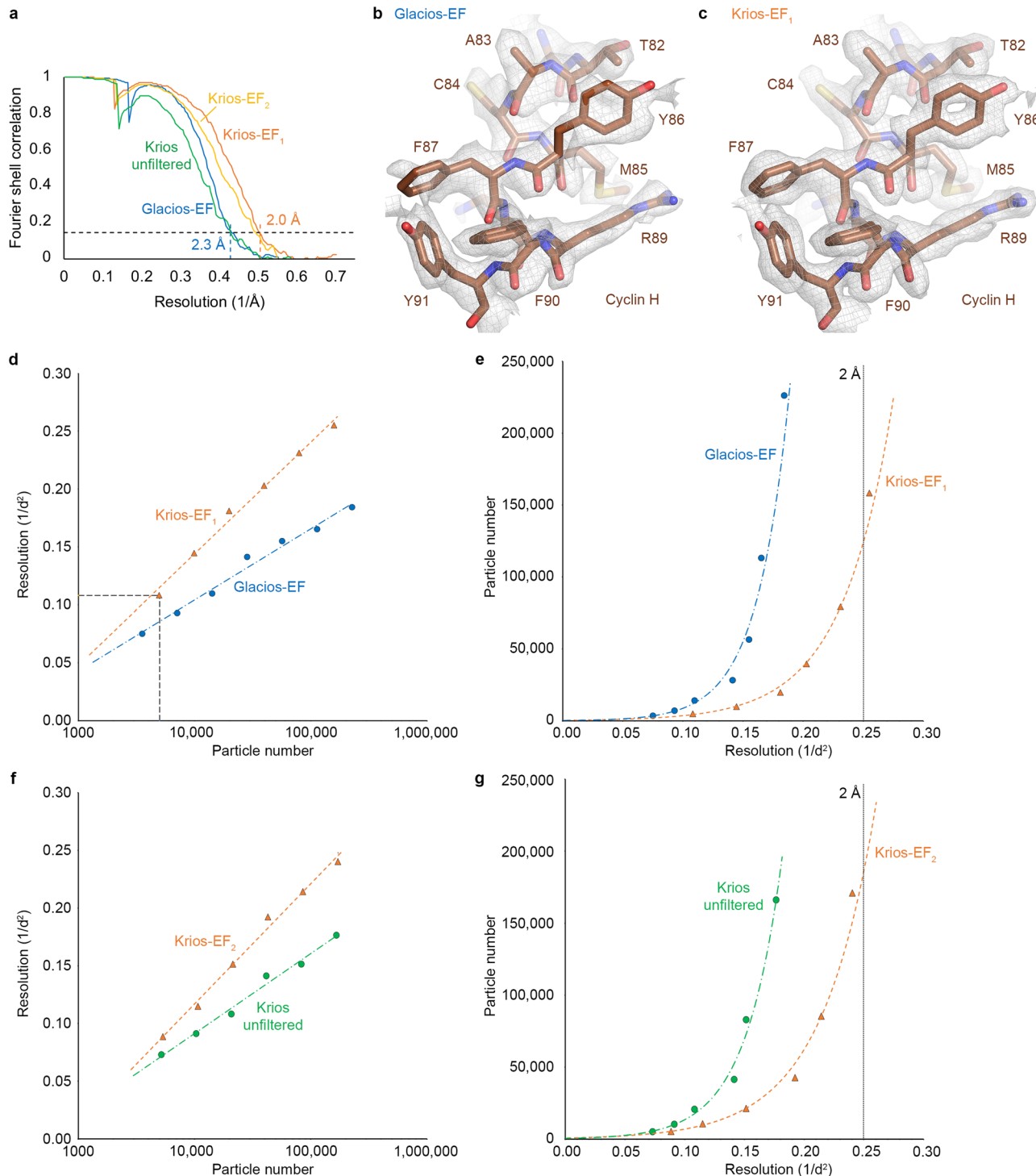

**Fig. 3 | Glacios vs. Krios resolution comparison and impact of energy filtration on the resolution obtained. a** FSC curves for the Glacios and Krios cryo-EM reconstructions (EF = energy filtered). Comparison between cryo-EM maps from Glacios-EF (**b**) and Krios-EF₁ (**c**) data. A segment of cyclin H is shown as an example. **d** Henderson-Rosenthal plot comparing Krios-EF₁ and Glacios-EF data. The inverse squared resolution of data subsets is plotted against the logarithm of the particle number, resulting in a linear relationship. **e** Particle number plotted against inverse squared resolution. It is apparent that the Glacios data may not cross the 2 Å line even for very large amounts of data. **f, g** As **d** and **e**, but comparing the reconstructions obtained from the energy-filtered Krios setup (dataset Krios-EF₂) against the results obtained after retracting the slit from the beam path (Krios unfiltered). Source data are provided as a Source Data file.

## Application of our cryo-EM workflow to structure determination of CAK-bound pyrazolopyrimidine-type inhibitors

The goal of the initial rapid sample screening phase was the identification of specimens exhibiting (i) good ice thickness, (ii) suitable particle density, (iii) suitable particle orientation distribution, (iv) presence of inhibitor, and (v) promising resolution from on-the-fly

processing of a dataset of a defined size during collection. Using our CAK-inhibitor complexes, we were able to satisfy these requirements using only 1 h of data collection time per dataset (Fig. 2b, c), thereby determining 26 structures confirming the presence of 12 different bound inhibitors from 13 samples, most of them at approximately 3.5–4.0 Å resolution (Supplementary Figs. 3, 4, 5a, Supplementary

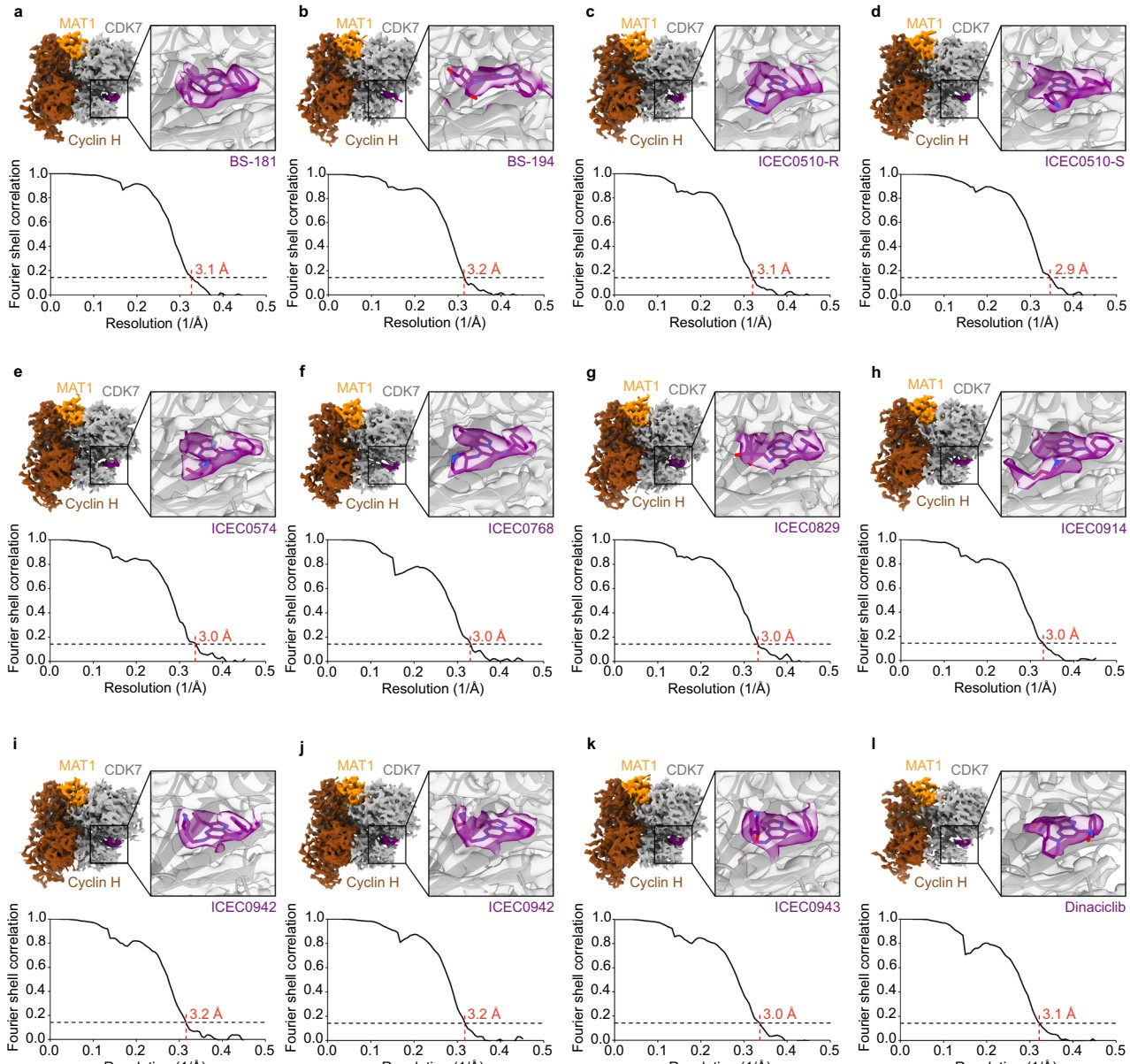

**Fig. 4 | Results of 4-h Glacios screening.** Overview of 4-h Glacios dataset results. Top panels show a view of the 3D reconstruction (CDK7 gray, cyclin H brown, MAT1 orange) and a close-up view of the model and density for bound inhibitors (purple). Bottom panels show the resolution according to the FSC = 0.143 threshold. **a** BS-181. **b** BS-194, a compound selective for CDK2. **c** ICEC510-R. **d** ICEC510-S. **e** ICEC0574. **f** ICEC0768. **g** ICEC0829. **h** ICEC0914. **i, j** The clinical inhibitor ICEC0942. **k** ICEC0943, the enantiomer of ICEC0942. **l** Dinaciclib, a compound selective for CDKs 1, 2, 5, and 9. Source data are provided as a Source Data file.

Table 3). All data processing at this stage was performed using cryoSPARC live[34], enabling rapid visualization of the results during data collection. To enable a quantitative assessment of map and model quality and allow comparisons with higher-resolution structures (see below), we plotted the Q-scores of map-model fits for these datasets (Supplementary Fig. 5b–d).

We subsequently performed intermediate-resolution structure determination using the 200 kV Glacios 2 setup on selected grids, aiming to produce cryo-EM maps suitable for preliminary interpretation and approximate inhibitor docking. To this end, we extended the data collection time to 4 h and achieved approximately 3 Å resolution for most datasets (Figs. 2d, 4, Supplementary Fig. 5a, Supplementary Table 4), which led to a substantial increase in the average Q-score for both the fitted protein and ligands (Supplementary Fig. 5b–d). We applied a combination of cryoSPARC and RELION for processing of these datasets (see "Methods" for details). The complete processing of more than a dozen datasets was feasible in approximately one day after the end of data collection on a single 4-GPU server. Parallelization of this step across multiple servers to reduce this time further will be straightforward if required. These acquisition sessions yielded 12 structures at around 3 Å resolution (Fig. 4), in which inhibitors can be oriented and refined into the density and major conformational differences in large substituents are visible. Models derived from these reconstructions may also be used as starting points for molecular dynamics simulations or other computational methods, an option that we did not further pursue at this stage because we opted to collect high-resolution data for these complexes.

Aiming to resolve CAK-bound inhibitors at high resolution to provide highly accurate molecular models and identify water molecules that may contribute to inhibitor binding and specificity, we

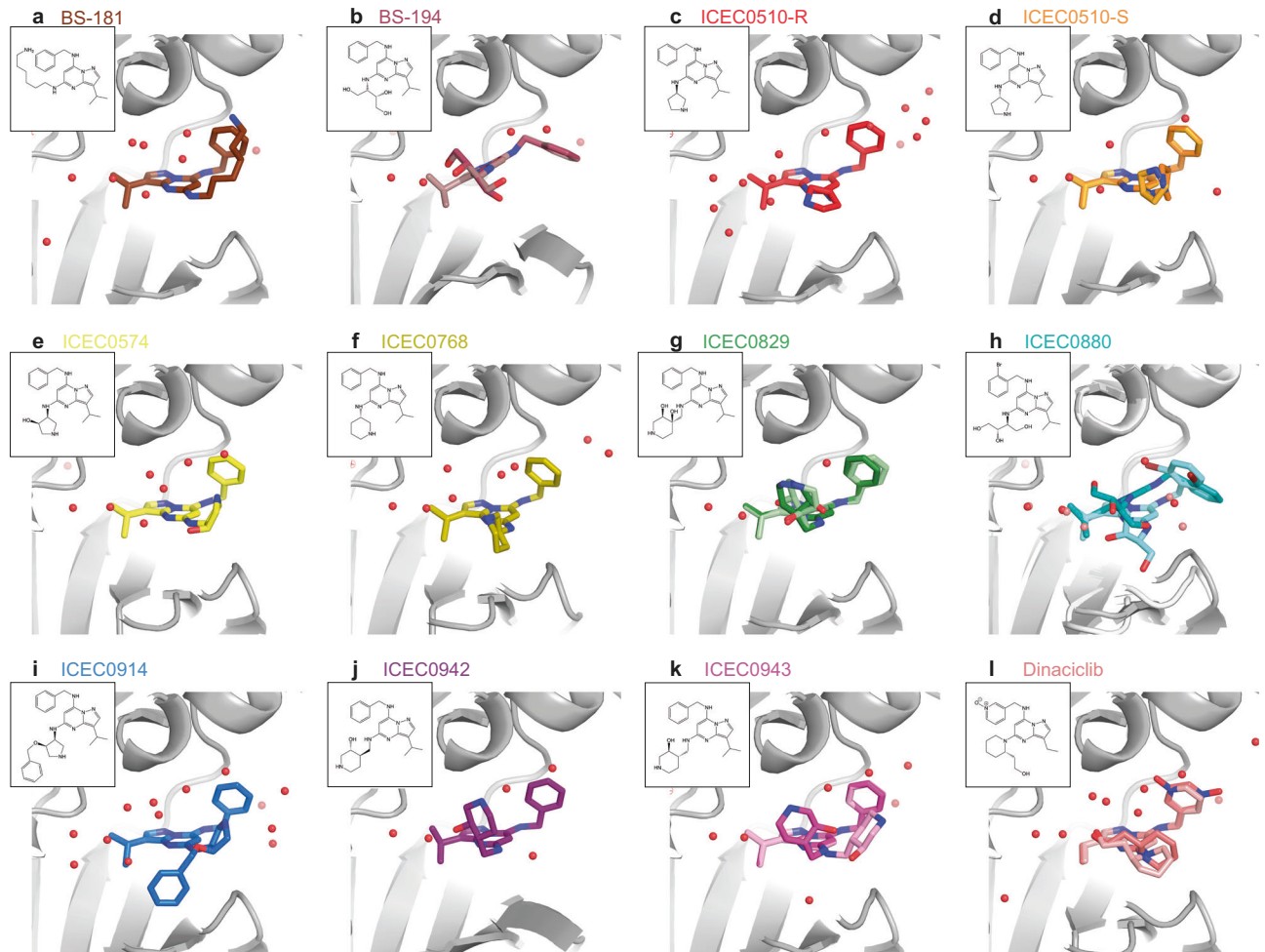

**Fig. 5 | Structures of pyrazolopyrimidine-type inhibitors bound to CDK7.** CDK7 is shown in gray, waters in red, ligands in color. Insets show the chemical structures of the corresponding inhibitor. **a** BS-181. **b** BS-194, a compound selective for CDK2 (two conformers with minor differences). **c** ICEC510-R. **d** ICEC510-S (two conformers). **e** ICEC0574 (two conformers). **f** ICEC0768. **g** ICEC0829 (two enantiomers). **h** ICEC0880 (two different positions correlated with conformational changes in CDK7). **i** ICEC0914. **j** The clinical inhibitor ICEC0942. **k** ICEC0943, the enantiomer of ICEC0942 (two conformers). **l** Dinaciclib (two conformers).

used the 300 kV Krios G4 setup for high-resolution data collection (Fig. 2b, e). Data collections lasted for approximately 10 h and yielded roughly 5,000 micrographs for each sample. These data were initially processed using cryoSPARC live and cryoSPARC[34] and final classifications, Bayesian polishing[35], CTF refinement[36] and 3D refinement were performed in RELION[37]. In contrast to the rapid screening and intermediate-resolution structure determination phases, the data processing effort for high-resolution reconstruction was substantial and extended substantially beyond the end of the data collection.

The large volume of CAK data acquired using this workflow (approximately 5000 micrographs per dataset for 12 inhibitors) allowed us to explore the resolution limits for our specimen and instrument by combining 6 fully processed datasets that had previously reached 1.8–1.9 Å resolution individually (see below). From 2,060,503 particles, we obtained a CAK reconstruction with averaged inhibitor density at a resolution of 1.7 Å after CTF refinement and an additional round of particle polishing (Supplementary Fig. 6a–c). Given the large volume of data that entered these computations, further progress towards atomic-resolution structure determination of the CAK is likely to depend on new specimen preparation methods, such as the introduction of nanobody derivatives to improve alignability of the particle images[38], or on new instrumentation (Supplementary Fig. 6d, e).

## High-resolution structures of CAK-bound pyrazolopyrimidine inhibitor complexes

Our high-throughput screening and collection workflow enabled us to efficiently visualize 12 CAK-inhibitor complexes at 1.8-2.2 Å resolution (Figs. 5a–l, 6a–f, Supplementary Figs. 5a, 7–10, Supplementary Tables 1, 5–11). The cryo-EM maps typically show well-resolved density for the pyrazolopyrimidine core and the benzylamine-derived substituents ($R_2$ position) of the bound inhibitors, and less-well resolved density for the variable substituents at the $R_1$ position, indicating structural heterogeneity in the bound compounds (e.g. Figs. 5g, h, k, l, 6c, d, f, Supplementary Figs. 8–10). The Q-scores for the protein and bound ligands are again improved compared to the 4-hour screening datasets, with average Q-scores reaching a value of approximately 0.8 in most structures (Supplementary Fig. 5b–d).

The bound ICEC-series compounds exploit hydrogen bonding interactions with the backbone amide and carbonyl of CDK7 residue M94 in the hinge region that are shared between all compounds, leading to a relatively well-preserved positioning of the pyrazolopyrimidine core within the CDK7 active site (Fig. 5a–l). These interactions are also exploited by other hinge-binding kinase inhibitors[10], including the CDK7 inhibitors THZ1 and LDC4297 (see above). Further interactions shared among the analyzed pyrazolopyrimidines are formed by the aliphatic substituent (ethyl for $R_3$ = H in dinaciclib and isopropyl for $R_3$ = $CH_3$ in all other inhibitors) that protrudes into a hydrophobic

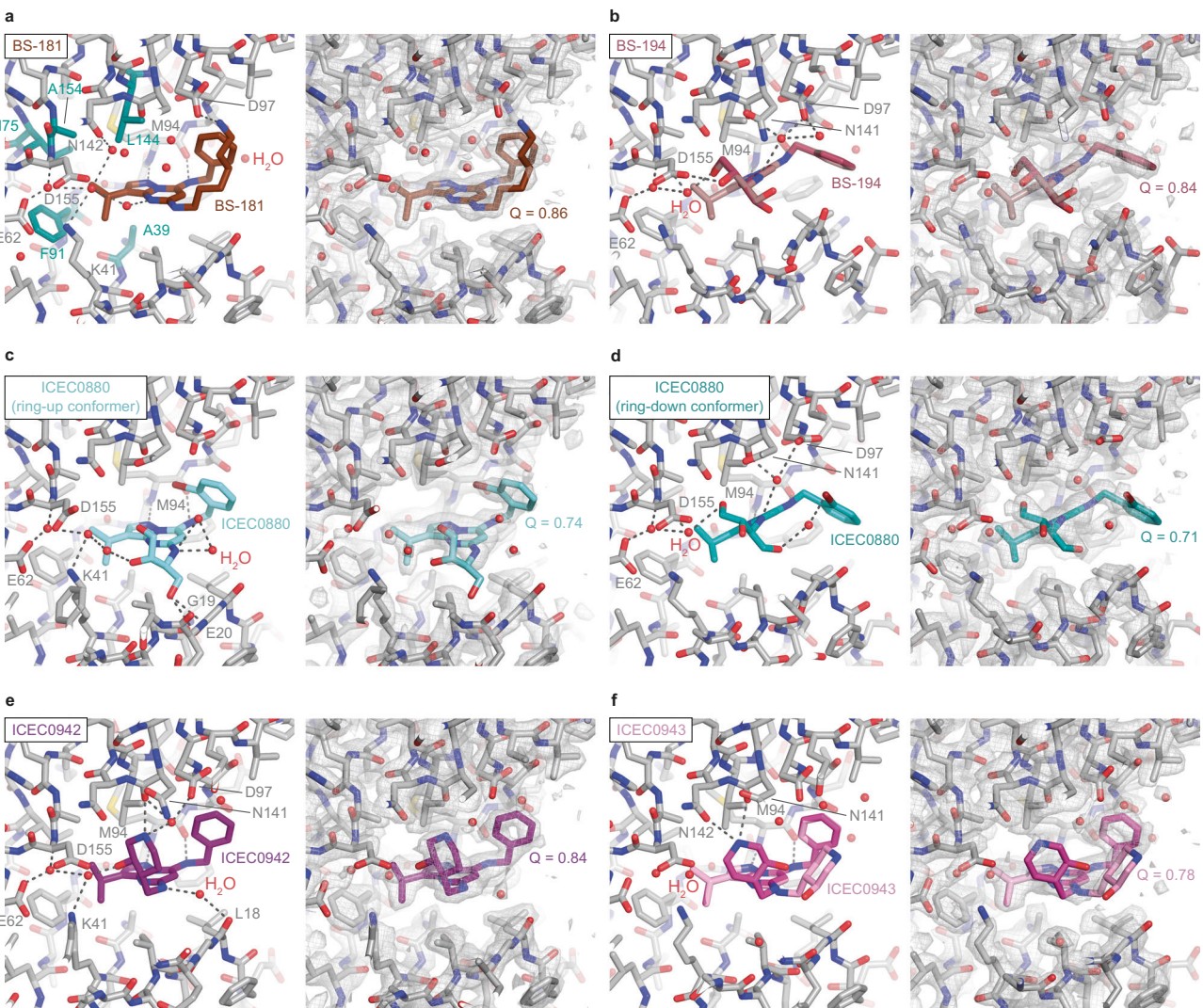

**Fig. 6 | Detailed analysis of selected CDK7-inhibitor complexes.** Hydrogen bonding interactions are indicated in the left sub-panel and the fit to the map including evaluation by Q-score is shown in the right sub-panel for each inhibitor. **a** BS-181, a highly CDK7-selective compound, assumes a ring-up conformation. Hydrophobic residues surrounding the R$_3$ substituent of the inhibitor (see text for details) are shown in teal. **b** BS-194, a highly CDK2-selective compound, assumes a ring-down conformation. **c, d** ICEC0880, a BS-194 derivative with an additional bromine atom at the benzylamine group, assumes both ring-up (**c**) and ring-down (**d**) conformations. **e** ICEC0942 engages in an extended network of water interactions that may be responsible for its high selectivity. **f** ICEC0943, the enantiomer of ICEC0942, appears to be unable to engage in the same water-mediated interactions, and its hydroxypiperidine substituent is found in two conformations.

cavity within the CDK7 active site formed by A39, I75, L144, A154, and the gatekeeper residue F91 (Fig. 6a). The R$_1$ substituent is the structurally most diverse among the inhibitors tested and engages in a variety of interactions that are specific to each inhibitor. For example, the protonated primary amine in BS-181 (Fig. 6a) forms a salt bridge with CDK7 residue D97, while the trihydroxybutyl substituent in BS-194 engages in a hydrogen bonding network that involves bound water molecules in the CDK7 active site cavity as well as CDK7 residues E62 and D155 (Fig. 6b). A more detailed analysis of the interactions and conformational dynamics of the R$_1$ and R$_2$ substituents (Fig. 6c–f) and their contributions to inhibitor selectivity will be provided in the following paragraphs.

**Structural insight into ICEC0942 selectivity**

The high-resolution analysis of the CAK-ICEC0942 complex (Fig. 6e) extends and further defines our previous results, in which the orientation of the hydroxypiperidine substituent at the R$_1$ position was ambiguous[14]. Our high-resolution density supports an orientation of the six-membered piperidine ring in which the hydroxy group points

into the active site cavity (Fig. 6e). This allows the hydroxy group to participate in a network of hydrogen bonding interactions involving two water molecules, the side chains of CDK7 residues K41 and E62, and the backbone amide of residue D155 (Fig. 6e, Supplementary Fig. 11a). Two additional water molecules are within hydrogen bonding distance of the protonated secondary amine in the hydroxypiperidine substituent and the exocyclic amino group connecting the R$_1$ substituent to the pyrazolopyrimidine core, respectively, and mediate interactions between the inhibitor and residues lining the CDK7 active site pocket (Fig. 6e). The protonated secondary amine of the hydroxypiperidine moiety is additionally hydrogen bonded to the backbone carbonyl of CDK7 residue N141 (Fig. 6e). It is worth noting that the equivalent group in the structurally related inhibitor CT7030 (see below) is slightly repositioned, by less than 1 Å, which additionally brings it into hydrogen bonding distance with the side chain of CDK7 residue N142. It is therefore possible that some conformations of ICEC0942 can access this interaction as well.

The hydroxy group on the ICEC0942 hydroxypiperidine substituent appears to be an important modulator of inhibitor binding,

likely due to its ability to participate in the hydrogen bonding networks outlined above. In agreement with this rationale, the interactions of the ICEC0942 hydroxypiperidine substituent in the CDK2 off-target complex are different. Here, the hydroxy group points out of the active site cavity and interacts with a water molecule that is hydrogen bonded to a backbone amide and carbonyl (Supplementary Fig. 11b). The hydroxypiperidine secondary amine is hydrogen bonded to the side chain of CDK2 residue N132, but positioned more than 4 Å away from the carbonyl oxygen of residue Q131 (the equivalents of CDK7 N142 and N141, respectively). Additionally, the benzylamine substituent assumes the ring-down conformation, likely due to small conformational differences of the hinge region and a leucine (CDK7) to isoleucine (CDK2) amino acid substitution in the N-terminal kinase lobe (Supplementary Fig. 11c, d). The side chain of CDK2 K89 (Supplementary Fig. 11b, d) could potentially interfere with the ring-up conformation, but the density for this side chain in the X-ray crystal structure (PDB ID 5JQ5)[16] is comparably weak, indicating that this side chain is partially flexible.

The importance of a specific configuration of the hydroxypiperidine group for CDK7 binding is further emphasized by the ICEC0942 enantiomer ICEC0943 (Fig. 2a). This compound exhibits poorly defined density for the entire hydroxypiperidine ring, suggestive of two different orientations (Fig. 6f). This inability to form the same stabilizing interactions in the CDK7 active site explains why ICEC0943 binds CDK7 with much reduced affinity, as reported previously[16].

### Further insights into CDK7 inhibitor selectivity

Our structural analysis yielded additional insight into inhibitor selectivity. BS-194 is the only inhibitor for which we observed a homogeneous ring-down conformation of the benzylamine substituent at the $R_2$ position (Figs. 5b, 6b). We observed previously that ICEC0942 assumes a ring-up conformation bound to CDK7 and a ring-down conformation bound to CDK2 (Fig. 7a), suggesting a link to target selectivity[14]. Comparisons of our data with existing structures show that dinaciclib[39] and ICEC0943[16] show a similar conformational switch between the CDK7 and CDK2-bound complexes, while BS-194 assumes a homogeneous ring-down conformation in complex with both kinases (Fig. 7b–d). Interestingly, BS-194 is strongly selective for CDK2 over CDK7, while the related derivative BS-181, which is selective for CDK7 over CDK2 (Supplementary Table 2)[30,31], exhibits a ring-up conformation bound to CDK7 (Figs. 6a, 7e). These observations support our hypothesis that this conformational change is due to properties of the respective CDK active sites, and that the ability of an inhibitor to accommodate this change may be connected to target preference, providing a possible secondary mechanism of the CDK7 selectivity of ICEC0942, in addition to the interactions formed by the hydroxypiperidine group (see above). The conformational change in the benzylamine ring is coupled to a shift of the pyrazolopyrimidine core of the inhibitors, which is observed for multiple compounds in CDK7 and CDK2 (Fig. 7a–f).

In contrast to the other inhibitors, which exhibit clear ring-up or ring-down conformations, ICEC0880 exhibits both conformations when bound to CDK7 (Figs. 5h, 6c, d, 7e, f). Because these inhibitor binding modes coincide with movements of the β-sheet in the N-terminal kinase lobe of CDK7, it was possible to computationally separate the cryo-EM data into distinct classes and directly visualize and compare them (Figs. 5h, 6c, d). Given the small molecular weight of the inhibitor, which produces only a minute signal in the raw particle images, this would likely have been impossible in the absence of correlated motions in the protein.

It is worth noting that both BS-194 and its derivative ICEC0880 contain a 1,2,4-trihydroxybutyl substituent in the $R_1$ position. It is possible that this substituent contributes to the observed effect by promoting a conformation of the pyrazolopyrimidine core that is conducive to the ring-down conformation and reduced CDK7 selectivity (or increased CDK2 selectivity), which is counteracted by the bulky exocyclic bromine at the benzylamine substituent of the ICEC0880. This added steric constraint might destabilize binding in the ring-down conformation, as exhibited by the shift of the entire pyrazolopyrimidine core in the two ICEC0880 structures. Notably, LDC4297, which is highly specific for CDK7, carries an even more bulky substituent at its ring-up benzylamine group (Supplementary Figs. 1b, 2e), and the recently reported CDK7-selective compound LGR6768[40] exploits a bulky biphenyl substituent at this position.

### The inhibitor CT7030 shows increased CDK7 selectivity

To directly test the idea that substituents at the benzylamine group could impinge on target selectivity, we performed in vitro kinase assays and structure determination of an inhibitor that is based on the ICEC0942 scaffold but harbors an *ortho*-methoxy substituent at the benzylamine group (CT7030, Fig. 8a). The cryo-EM structure of the CAK-CT7030 complex confirms that the inhibitor assumes the expected ring-up conformation (Fig. 8b, Supplementary Fig. 12), and in vitro kinase inhibition assays confirmed an increase in selectivity for CDK7 over CDK2 to more than 200× (Fig. 8c–e), supporting the validity of our cryo-EM-based mechanistic hypothesis.

## Discussion

We report a series of structures of the human CAK in free, nucleotide-bound, and small-molecule inhibitor-bound states, 14 of them at 2 Å resolution or better. Notably, we obtained these results using a molecular complex that is the target of actively ongoing drug discovery programs and clinical trials. Before our efforts, the structure of only one <100 kDa protein complex had been determined at better than 2 Å resolution by cryo-EM—that of streptavidin (e.g. EMD-31083)—and none for a medically relevant target or a protein complex lacking symmetry in this size range. We have now demonstrated that current cryo-EM technology is suitable for routine 2 Å structure determination of small protein complexes from less than one day of data collection on a high-end system, opening the door to high-throughput, high-resolution cryo-EM even for nominally difficult targets[18]. Additionally, our screening approaches demonstrate the feasibility of the determination of more than a dozen 3.5–4.0 Å-structures a day, and several 3 Å structures a day, using a 200 kV microscope. Using a 300 kV instrument for screening harbors the potential for even higher sample throughput at the same resolution, or improved resolution at constant sample throughput. These results and the methodology outlined in this work will facilitate the application of cryo-EM to iterative structure-based drug design efforts, which are of critical importance to address global health challenges.

Our data show that the hydroxypiperidine substituent of the clinical inhibitor ICEC0942 engages in favorable interactions in the CDK7 active site, which results in the high affinity and selectivity of the compound. Our data additionally support a mechanistic hypothesis for a secondary mechanism of CDK7 selectivity in pyrazolopyrimidine-based CDK inhibitors, specifically that small structural differences in the nucleotide-binding pockets in CDK7 and CDK2 induce a preference for different conformations of the benzylamine substituent that forms part of the scaffold of the inhibitors we analyzed, and that chemical modifications that favor one conformation over the other may be used to improve kinase selectivity of these compounds. These findings will pave the way for future structure-guided CDK7 inhibitor design. Furthermore, they provide a proof of principle for the benefit of the application of cryo-EM in structure-based CDK7 inhibitor design.

## Methods
### Synthesis of compounds
All CDK inhibitors were synthesized through the course of prior studies using methods described[30,31]. In brief, all compounds were

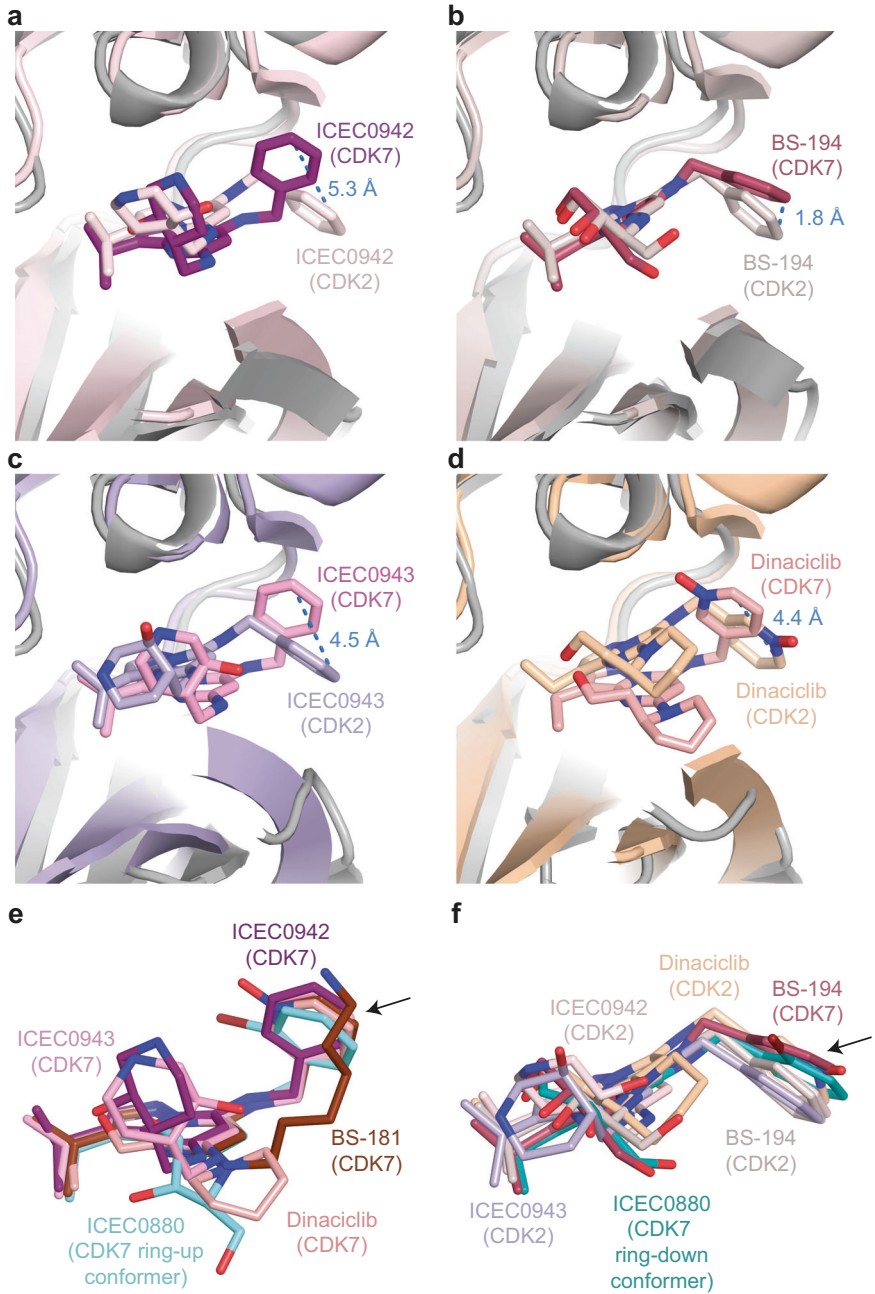

**Fig. 7 | Analysis of high-resolution inhibitor structures. a** Comparison of ICEC0942 bound to CDK7 (this work) and CDK2 (PDB ID 5JQ5)[16]. The conformation of the benzylamine group and the placement of the inhibitor core differ between the two structures. The displacement distance of the most distal carbon atom of the phenyl ring in the benzylamine substituent is indicated. **b** Comparison of BS-194 bound to CDK7 (this work) and CDK2 (PDB ID 3NS9)[31]. **c** Comparison of ICEC0943 bound to CDK7 (this work) and CDK2 (PDB ID 5JQ8)[16]. **d** Comparison of dinaciclib bound to CDK7 (this work) and CDK2 (PDB ID 4KD1)[39]. For clarity, only one conformer for the CDK7-bound compounds is shown in (**b**–**d**). **e** Superposition of ICEC0942, BS-181, ICEC0943, dinaciclib, and the ring-up ICEC0880 conformer, all bound to CDK7. **f** Superposition of ICEC0942, BS-194, ICEC0943, and dinaciclib, all bound to CDK2, with BS-194 and the ring-down conformer of ICEC0880 bound to CDK7. The flipping ring system is indicated with an arrow.

prepared from dichloropyrazolo[1,5-*a*]pyrimidine by sequential selective substitution of the chlorides by amines. Firstly, the $R_2$ substituent was added by nucleophilic aromatic substitution using General Procedure A, followed by protection of the NH using General Procedure B. Protection of the benzylic amine as a carbamate was essential for successful amination of the second ($R_1$) position. The $R_1$ substituent was added using a palladium-catalyzed Buchwald-Hartwig reaction, according to General Procedure C, followed by deprotection (General Procedure D).

**General Procedure A.** Dichloropyrazolo[1,5-*a*]pyrimidine (1.0 equiv) and the $R_2$ amine (2.0 equiv) in EtOH (0.13 M) were heated at reflux for 16 h. The reaction mixture was cooled to ambient temperature and concentrated *in vacuo*. Crystallization (EtOAc) or chromatography (EtOAc: hexanes) was used for purification.

**General Procedure B.** The relevant substrate (1.0 equiv), di-*tert*-butyl dicarbonate (1.3 equiv), and 4-dimethylaminopyridine (catalytic) in THF (0.14 M) were stirred for 20 h at ambient temperature. EtOAc was

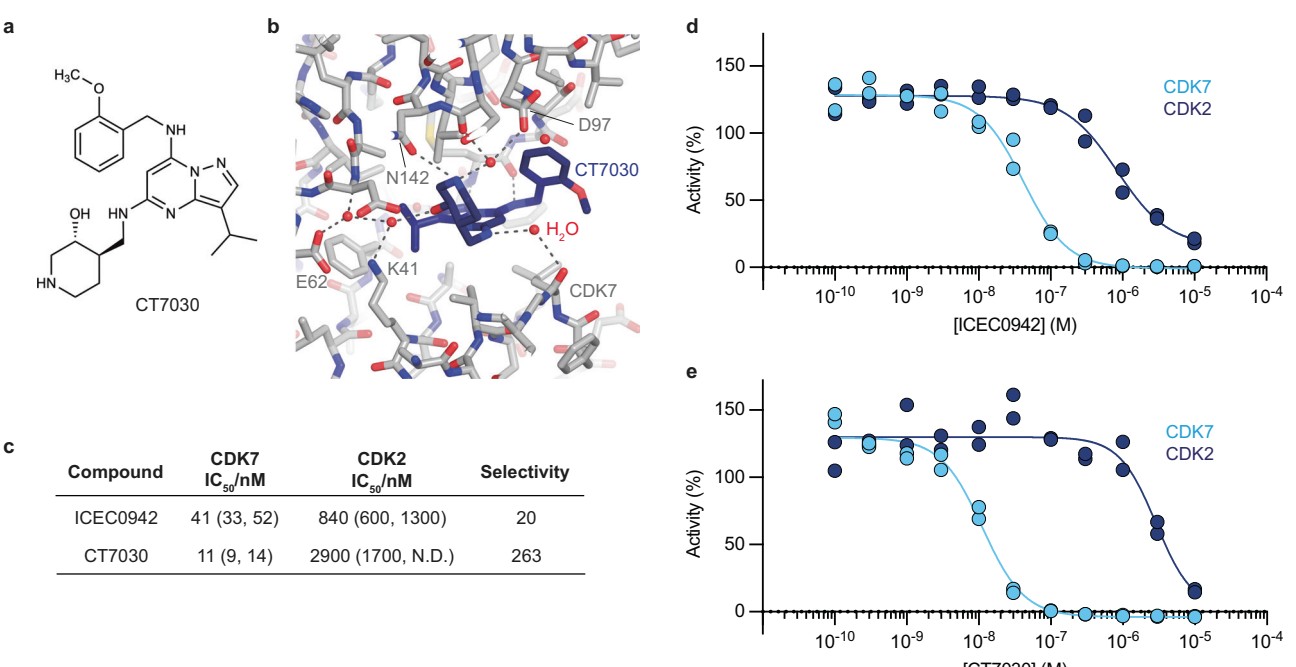

**Fig. 8 | The CDK7 inhibitor CT7030. a** Chemical structure of CT7030. **b** Cryo-EM structure of CAK-bound CT7030. **c** In vitro kinase inhibition data of CT7030 compared with ICEC0942. Numbers in brackets indicate 95% confidence intervals for the IC$_{50}$ values (ND not determined). **d, e** Plots of the data underlying the values in **a** ($n = 2$). Source data are provided as a Source Data file.

added, and the organic layer was washed with water (2×), saturated aqueous NaHCO$_3$, and dried (Na$_2$SO$_4$). The organic solvent was removed *in vacuo*. Chromatography on silica (EtOAc: hexanes) was used for purification.

**General Procedure C.** The relevant substrate (1.0 equiv) was reacted with tris(dibenzylideneacetone)dipalladium (0.05 equiv), 2,2′-bis(diphenylphosphino)-1,1′-binaphthalene (0.15 equiv), and sodium *tert*-butoxide (1.1 equiv) in PhMe (0.30 M). The mixture was stirred for 15 min. The corresponding R$_1$ amine (1.1 equiv) was added, and the mixture heated to 95 °C for 16 h and cooled and diluted with EtOAc. The organic phase was washed with water (3X), brine, dried (MgSO$_4$), and concentrated *in vacuo*. Chromatography on silica (EtOAc: hexanes) was used for purification.

**General Procedure D.** AcCl was dissolved in MeOH (5.0 M) with stirring at 0 °C. After 15 min, the solution was warmed to room temperature and stirred for a further 15 min. The relevant substrate was added, and the mixture stirred at ambient temperature for 3 h and concentrated *in vacuo*. The residue was dissolved in CH$_2$Cl$_2$, washed with saturated aqueous sodium bicarbonate, dried (Na$_2$SO$_4$), and concentrated *in vacuo*. Crystallization (MeCN or MeOH) or chromatography (CH$_2$Cl$_2$: MeOH) was used for purification.

### Characterization of ICEC0510 and ICEC0510-R/S

m.p. (MeOH) = 200–230 °C. IR (neat) $v_{max}$ 3316, 2911, 2718, 2428, 1636, 1588, 1579, 1553, 1495, 1445, 1418, 1356, 1235, 1170, 765, 698 cm$^{-1}$. 1H NMR (400 MHz, DMSO-D): δ. 9.16 (s, 2H, NH+), 7.96 (t, J = 6.5 Hz, 1H, NH), 7.66 (s, 1H), 7.34−7.31 (m, 4H), 7.27−7.22 (m, 1H), 6.99 (d, J = 5.2 Hz, 1H, NH), 5.09 (s, 1H), 4.46 (d, J = 6.5 Hz, 2H), 4.41−4.34 (m, 1H), 3.42 (dd, J = 11.8 Hz and J = 6.5 Hz, 1H), 3.33−3.17 (m, 2H), 3.04 (dd, J = 11.8 Hz and J = 4.8 Hz, 1H), 2.98 (heptuplet, J = 6.9 Hz, 1H), 2.21- 2.12 (m, 1H), 1.91−1.82 (m, 1H), 1.265 (d, J = 6.9 Hz, 3H), 1.262 (d, J = 6.9 Hz, 3H). 13 C NMR (125 MHz, DMSO-D): δ 155.6 (s), 146.2 (s), 145.0 (s), 139.8 (d), 138.3 (s), 128.4 6(d, 2C), 126.9 (d), 126.6 (d, 2C), 111.0 (s), 72.5 (d), 49.8 (t),

44.5 (t), 43.5 (t), 29.8 (t), 23.4(d), 23.1 (q), 23.0 (q). Elemental Analysis: Calculated for C$_{20}$H$_{27}$ClN$_6$: C, 62.08; H, 7.03; N, 21.72. Found: C, 62.18; H, 6.98; N, 21.68.

Racemic ICEC0510 was separated into *R* and *S* enantiomers using chiral HPLC, with [α]$_D$ = −6.0 (c 0.110 in MeOH) and [α]$_D$ = 8.5 (c 0.095 in MeOH) for ICEC0510-R and -S, respectively.

### Characterization of ICEC0914

1H NMR (600 MHz, MeOD) δ 7.69 (s, 1H), 7.44−7.41 (m, 2H), 7.38−7.34 (m, 2H), 7.30−7.27 (m, 1H), 7.23 (dd, J = 7.6, 1.8 Hz, 2H), 7.19−7.13 (m, 3H), 4.72 - 4.67 (m, 1H), 4.64 (d, J = 12.0 Hz, 1H), 4.60 (br s, NH), 4.59 (s, 2H), 4.50 (d, J = 12.0 Hz, 1H), 4.30−4.38 (m, 1H), 3.68−3.58 (m, 1H), 3.55−3.49 (m,1H), 3.45−3.38 (m,1H), 3.25−3.18 (m, 1H), 3.04 (heptuplet, J = 6.9 Hz, 1H), 1.38-1.25 (m, 6H). 1H obscured by solvent peak. LC/MS (ES + ): 1.90 min. Found: 457.2716. Calculated for C$_{27}$H$_{32}$N$_6$O (M + H$^+$): 457.2710.

The amount of ICEC0914 available was insufficient for further characterization. However, inhibitor characterization using LC−MS (see below and Supplementary Table 12) and the presence of all expected chemical groups in our cryo-EM map of the compound-bound complex further support the identity of the compound.

### Characterization of ICEC0768

m.p. = 175−180 °C. [α]$_D$ −20.2 (*c* 0.47, MeOH). IR (neat) $v_{max}$ 3293, 1635, 1575, 1441, 1360, 1219, 1173, 1063, 748, 699 cm$^{-1}$. 1H NMR (400 MHz, MeOD): δ 7.73 (s, 1H), 7.43−7.36 (m, 4H), 7.31−7.28 (m, 1H), 5.29 (br s, 1H), 4.61 (br s, 2H), 4.26−4.15 (m, 1H), 3.61−3.57 (m, 1H), 3.33−3.26 (m, 1H), 3.12 (heptuplet, J = 6.9 Hz, 1H), 3.07−3.00 (m, 2H), 2.12−2.05 (m, 2H), 1.92−1.81 (m, 1H), 1.75−1.64 (m, 1H), 1.340 (d, J = 6.9 Hz, 3H), 1.336 (d, J = 6.9 Hz, 3H). 13C NMR (125 MHz, MeOD): Due to low signal intensity, some carbons could not be seen. δ 141.8 (d), 138.9 (s), 129.8 (d, 2C), 128.5 (d, C), 128.0 (d, 2C), 113.6 (s), 48.5 (t), 46.6 (t), 46.2 (d, low signal intensity), 45.0 (t), 29.2 (t), 24.8 (d), 23.8 (q), 23.7 (q), 21.9 (t). LC/MS (ES+): 3.71 min. Found: 365.2442. Calculated for C$_{21}$H$_{29}$N$_6$ (M + H$^+$): 365.2454.

## Characterization of ICEC0880

IR $v_{max}$/cm$^{-1}$ (film) 3275 (w), 1637 (s), 1576 (m), 1553 (s), 1424 (m); 1H NMR δH (400 MHz, MeOD) 7.68 (1H, s), 7.62 (1H, dd, J 7.9, 1.1 Hz), 7.38 (1H, dd, J 7.7, 1.5 Hz), 7.32 (1H, td, J 7.5, 1.1 Hz), 7.20 (1H, td, J 7.7, 1.9 Hz), 5.20 (1H, s), 4.60 (2H, s), 4.03 (1H, m), 3.84 (1H, dd, J 11.2, 3.8 Hz), 3.78 (1H, dd, J 11.1, 5.9 Hz), 3.54 (3H, m), 3.02 (1H, sp, J 6.9 Hz), 1.29 (3H, d, J 6.9 Hz),

1.26 (3H, d, J 6.9 Hz); 13C NMR δC (100 MHz, MeOD) 159.0, 148.2, 146.0, 141.7, 137.6, 134.0, 130.3, 129.5, 129.0, 123.9, 113.6, 74.3, 73.5, 64.5, 63.1, 55.8, 47.0, 24.7, 24.0, 23.6; m/z HRMS (ES+): (M + H) + 464.1295, $C_{20}H_{27}BrN_5O_3$ requires 464.1297.

## Characterization of CT7030

1H NMR (400 MHz, DMSO) δ 7.64 (s, 1H), 7.57 (t, J = 6.4 Hz, 1H), 7.26 (td, J = 7.8, 1.7 Hz, 1H), 7.16 (dd, J = 7.5, 1.7 Hz, 1H), 7.03 (dd, J = 8.4, 1.1 Hz, 1H), 6.90 (td, J = 7.5, 1.1 Hz, 1H), 6.71 (t, J = 6.1 Hz, 1H), 5.38 (s, 1H), 5.12 (s, 1H), 4.41 (d, J = 6.4 Hz, 2H), 3.87 (s, 3H), 3.52 (m, 1H), 3.20 (m, 1H), 3.06 (m, 1H), 2.95 (m, 2H), 2.83 (m, 1H), 2.35 (m, 1H), 2.21 (dd, J = 11.6, 10.0 Hz, 1H), 1.58 (m, 1H), 1.36 (m, 1H), 1.25 (d, J = 4.7 Hz, 3H), 1.23 (d, J = 4.7 Hz, 3H), 1.16 (m, 1H). 1H was not observed.

The amount of CT7030 available was insufficient for further characterization. However, inhibitor characterization using LC-MS (see below and Supplementary Table 12) and the presence of all expected chemical groups in our cryo-EM map of the compound-bound complex further support the identity of the compound.

## Compounds used for structural and enzyme inhibition studies

ATPγS was purchased from Sigma Aldrich; THZ1 was purchased from EMD Millipore; LDC4297 and dinaciclib were purchased from MedChemExpress (distributed by Insight Biotechnology, Wembley, UK). ICEC-series pyrazolopyrimidine compounds were synthesized and characterized during the course of the drug discovery program that led to the discovery of ICEC0942[5,30–33,41]. Further characterization data for ICEC0510-R and S enantiomers, ICEC0768, ICEC0880, ICEC0914, and CT7030 are provided above. After retrieval from cold storage, integrity of ICEC-series compounds was verified using UV-vis spectroscopy and liquid chromatography/mass spectrometry (see below for detailed methods). The results of this characterization are summarized in Supplementary Table 12.

## Verification of inhibitor integrity and purity

LC/MS grade solvents, formic acid, or alternative eluent modifiers were purchased from VWR (Poole, UK) and Fisher (Loughborough, UK) unless otherwise stated. 10 μL of 1 mM DMSO solution of each compound was plated in a 384 Greiner (781280) well plate. 0.1 μL standard injections (with needle wash) of the sample were made onto a Phenomenex Kinetex C18 30 × 2.1 mm, 2.6u, 100A column (Phenomenex, Torrance, CA, USA). Chromatographic separation was carried out at 40 °C using an Agilent 1260 Infinity II series UPLC (Agilent, Santa Clara, USA) over a 4 min gradient elution (KNOWNS_AM190319.m) from 90:10 to 10:90 water:methanol (both modified with 0.1% formic acid) at a flow rate of 0.4 mL/min. UV–vis spectra were acquired at 254 nm on a 1260 Series diode array detector (Agilent, Santa Clara, USA).

The post column eluent flow from the diode array detector was split, with 90% sent to waste. The remainder was infused into a 6530 Series QtoF mass spectrometer fitted with an Agilent Jet Stream ESI source (Agilent, Santa Clara, USA). LC eluent and nebulizing gas was introduced into the grounded nebulizer with spray direction orthogonal to the capillary axis. A nozzle voltage of 0 V was applied to the charging electrode to generate a charged aerosol. The aerosol was dried by heated drying gas (10 L/min of nitrogen at 350 °C, 35 psi), producing ions by ESI. Ions entered the transfer capillary along which a potential difference of 4 kV was applied. The fragmenter voltage was set at 175 V and skimmer at 65 V. Signal was optimized by SWARM autotune. Profile mass spectrometry data was acquired in positive ionization mode over a scan range of m/z 190–650 (scan rate 4.0) with reference mass correction at m/z 622.02896 hexakis(2,2-difluoroethoxy)phosphazene.

Raw data were processed using Agilent MassHunter Qualitative Analysis B.07.00 (AutoQC.m). The "Find Compounds by Formula" algorithm was used to identify compounds and calculate the purity. The compound purity was calculated using the highest value of %UV (at 254 nm) or %TIC (total ion count from the mass spectrometer).

## Enzyme inhibition assays

In-vitro kinase assays comparing CT7030 and ICEC0942 activity were performed by the ADP-Glo method by Sygnature Discovery (Nottingham, UK) using in-house enzymes. Raw data were normalized using DMSO (negative) and staurosporine (positive) controls on the same plates. Data were analyzed in PRISM (versions 9 and 10; GraphPad Software, Inc.) as inhibitor concentration vs. normalized response with variable slope. Asymmetric confidence intervals were computed at a significance level of 95%. In-vitro kinase assays of inhibitors in Supplementary Table 2 reported in the literature were performed in ProQinase format[5,16], and we used the same assay, performed by Reaction Biology GmbH (Freiburg, Germany) for enzyme inhibition analysis reported in Supplementary Table 2 and Supplementary Fig. 13. ATP concentrations were set to the apparent $K_m$ of the respective enzyme.

## CAK expression, purification, and cryo-EM specimen preparation

For expression of all CAK variants used in this work, 438-series vectors encoding the multi-protein expression constructs[42] were transformed into EMBacY cells for preparation of bacmids[43] by isopropanol precipitation. Bacmid DNA was transfected into Sf9 (*Spodoptera frugiperda*) insect cells (Thermo Fisher Scientific, catalog number 11496015) using CellFectin II transfection reagent (Thermo Fisher Scientific). Baculoviruses were amplified for two rounds in Sf9 cells (one round in Sf9 and one round in Hi5 for CAK-MAT1Δ219). Subsequently, 15 mL of insect cell culture supernatant (50 mL of cell culture for CAK-MAT1Δ219) were used for infection of 1-L cultures of High5 (*Trichoplusia ni*) insect cells (Thermo Fisher Scientific, catalog number B85502). After 60–72 h of incubation, insect cells expressing protein complexes were harvested, frozen in liquid $N_2$, and stored at −80 °C for later use.

CAK was purified by nickel-affinity, strep-tactin, and gel filtration chromatography[15]. Cells were thawed and stirred during resuspension in lysis buffer (250 mM KCl, 40 mM HEPES-KOH pH 7.9, 5 mM MgCl$_2$, 5 mM β-mercaptoethanol, 10 mM imidazole and 10% (v/v) glycerol supplemented with protease inhibitors and DNaseI). After lysis by sonication, cell debris were pelleted by centrifugation at 18,000 rpm (approx. 26,000 × g) for 30 min at 4 °C in a JA-25.50 rotor (Beckman Coulter) and the cleared lysate was incubated with Ni-NTA superflow resin (Qiagen) for 30 min. The beads were washed using CAK purification buffer (250 mM KCl, 25 mM HEPES-KOH pH 7.9, 2 mM MgCl$_2$, 5 mM β-mercaptoethanol, and 10% (v/v) glycerol) supplemented with 25 mM imidazole. After elution using purification buffer supplemented with 300 mM imidazole, the eluted fractions were incubated with Strep-Tactin superflow plus resin (Qiagen) for 30–45 min. The beads were washed and purified protein was eluted using purification buffer supplemented with 10 mM desthiobiotin (Sigma-Aldrich). Affinity tags were cleaved using Tobacco Etch Virus (TEV) protease at room temperature for 2 h, and tags as well as His$_6$-tagged protease were removed using reverse-phase nickel affinity chromatography. Size exclusion chromatography of the fractions containing CAK was performed using a Superdex 200 10/300 GL column (GE Healthcare) immediately after the reverse-phase nickel affinity chromatography or on the next day in the case of CAK-MAT1Δ219, in which case the protein was snap-frozen in liquid nitrogen and stored at −80 °C over night. Peak fractions were pooled, concentrated to 2 mg/mL, and frozen in

liquid $N_2$ for storage at −80 °C. Most datasets used the CAK-MAT1Δ219 variant, in which the N-terminal 219 residues of MAT1 are replaced by an MBP-tag, which is cleaved by incubation with TEV protease to yield the catalytic module of CAK lacking the TFIIH-binding portion of MAT1[15]. The LDC4297 dataset additionally used a construct in which the N-terminus of CDK7 is fused to three YSPTSPS-repeats as a substrate mimic resembling the Pol II-CTD, which has however never been visualized in any density arising from data using this construct[15]. CAK covalently modified by THZ1 was prepared by incubating 2 µM of CAK-MAT1Δ219 with 5 µM THZ1 (EMD Millipore) in buffer devoid of reducing agent for 1.5 h at room temperature, followed by gel filtration on a Superdex 200 10/300 GL column (GE Healthcare) to remove excess inhibitor and residual DMSO.

Cryo-EM specimens were prepared on UltraAuFoil R1.2/1.3 holey gold grids (Quantifoil Microtools). For complex formation, CAK complex at approximately 2 mg/mL stock concentration was diluted 5−6× in sample buffer (20 mM HEPES-KOH pH 7.9, 200 mM KCl, 2 mM MgCl$_2$, 5 mM β-mercaptoethanol for most samples; no β-mercaptoethanol for CAK-ICEC0574, CAK-ICEC0829, CAK-ICEC0942, apo-CAK, and CAK-ATPγS) and incubated with 50 µM non-covalent inhibitor (dissolved at 50 mM concentration in 100% DMSO, except ICEC0829 and ICEC0942, which were dissolved in water; no added inhibitor in case of CAK-THZ1, which was prepared as a covalent adduct) or 2 mM ATPγS (Merck Life Science UK) at room temperature for 5 min. 4 µL of CAK complex were applied to a plasma cleaned grid (Tergeo plasma cleaner, PIE scientific) mounted in a Vitrobot Mark IV (Thermo Fisher Scientific) operated at 5 °C and 100% humidity, blotted for 1, 1.5, 1.5, or 2 sec (4 grids made for each complex) and plunged into liquid ethane at liquid $N_2$ temperature. After vitrification, grids were clipped into autogrid cartridges (Thermo Fisher Scientific) for use with Glacios and Krios autoloader systems.

### Initial screening

Initially, grids were coarsely screened on a Glacios cryo-transmission electron microscope (cryo-TEM) operated at 200 kV acceleration voltage and equipped with a Falcon 4 (later Falcon 4i) direct electron detector. Data were acquired in EER format at a pixel size of 0.5675 Å/pixel using a flux of 4−6 electrons. pixel$^{-1}$ s$^{-1}$ using aberration-free image shift (AFIS) to accelerate the data collection. Approximately 300-600 micrographs were processed in real time using cryoSPARC live[34] using 2× binning and with exposures fractionated into 60 frames for motion correction, which provided results suitable to judge overall sample quality and particle orientation distribution. This approach was later supplanted by the rapid-screening approach on a Glacios 2 electron microscope equipped with a Falcon 4i detector, a Selectris X energy filter, and fringe-free imaging capability enabling collection of 2−3 movies per hole and increased data collection rates. The latter data are shown in the "Results" section.

### Multigrid-screening and data processing

We used acquisition settings that were similar to our previously reported conditions for structure determination of the CAK[14,15], adjusted for the changes in hardware compared to prior experiments, resulting in 0.57 Å pixel size, 70 electrons. Å$^{-2}$ total exposure at 7−8 electrons. pixel$^{-1}$ s$^{-1}$ and EER format fractionated into 50 frames for screening runs and 70 frames for high-resolution runs (see below).

Data were collected using EPU-Multigrid (EPU version 2.14) on a 200 kV Glacios 2 microscope equipped with a Selectris X energy filter coupled with a Falcon 4i detector (Thermo Fisher Scientific). Data were collected using fringe-free imaging (FFI) and AFIS at a throughput of roughly 500 images/h. 2 grids each of 5 different samples and a dry Quantifoil Cu200 R2/2 or S106 cross-grating (11 grids total) were loaded into the autoloader for each EPU-Multigrid run. Within EPU, a new session queue was created, followed by sequentially loading grids to the stage for session set-up and grid square selection to give approximately 500 (for 1 h-screening runs) or 2000 (for 4-h data collection session) images after automatic ice-filter selection. Before starting data collection, two-fold astigmatism was corrected, and beam tilt was adjusted to the coma-free axis using the dry Quantifoil Cu200 R2/2 or S106 cross-grating. The Selectris X filter slit was then centered on the zero-loss peak with a slit width of 10 eV, and the filter tuned for isochromaticity, magnification and chromatic aberrations using Sherpa software (version 2.0; Thermo Fisher Scientific) over vacuum. The EPU Multigrid queue was then started, and each grid was automatically loaded sequentially onto the stage, whereupon grid squares previously selected were automatically brought to eucentric height and holes for data collection were automatically selected with ice filter settings. Images were collected at a nominal magnification of 205,000x with a resulting pixel size of 0.57 Å. Flux measured on the detector over vacuum at spot size 3 at parallel illumination conditions was 7.94 electrons· pixel$^{-1}$ s$^{-1}$ and each exposure was 2.85 s long to accumulate a total dose of 70 electrons. Å$^2$ per exposure. Images were collected at defocus values −0.8 µm, −1.0 µm and −1.2 µm. AFIS together with FFI and a 20 µm C2 aperture was employed to acquire 2 exposures per 1.2 µm ice hole and to accelerate data collection to approximately 500 per hour.

Rapid-screening and intermediate-resolution data were processed in cryoSPARC live (version 3.3.1)[34]. Movies were binned 2× during motion correction. Particles were picked using blob picker (elliptical blob, 80−110 Å diameter) and extracted in 160 × 160-pixel boxes. On-the-fly 2D classification and 3D refinement were performed using default parameters, with a previous reconstruction of CAK serving as an initial reference. Using a workstation with 4 RTX3090Ti GPUs (Nvidia Corporation), the software was able to match data collection rates.

The rapid screening workflow used cryoSPARC live processing exclusively. To improve interpretability of the maps in the intermediate resolution workflow, we added two steps after cryoSPARC live processing: (i) We performed an additional 2D classification of all live-picked particles (except obvious non-particles and artifacts) which led to recovery of more particles and clear classes for additional particle views. Subsequently (ii), we performed a 3D refinement, an alignment-free 3D classification, and a final refinement of the best class in RELION 4.0[37], providing the final reconstruction. The resolutions were typically almost unchanged (within 0.1 Å) compared to the cryoSPARC live output, but the RELION maps were cleaner. The cryoSPARC 2D classification took approximately 2 h per dataset, and the downstream processing in RELION was completed in approximately 1.5 h.

### High-resolution data collection

Data were collected on a Krios G4 cryo-TEM (Thermo Fisher Scientific) equipped with a cold-FEG operating at 300 kV acceleration voltage, a Selectris X energy filter, and a Falcon 4i direct electron detector. Data were acquired using EPU (version 3.1) at 0.57 Å pixel size, 70 electrons· Å$^{-2}$ total exposure at 7−8 electrons· pixel$^{-1}$ sec$^{-1}$ and in EER format, later fractionated into 70 frames during data processing. AFIS and FFI were employed to acquire 2-3 exposures per 1.2 µm ice hole and to accelerate data collection to approximately 600 movies/h by collecting data in holes within 12 µm of image-beam shift radius from the centered position. Approximately 5000 images were collected per grid for most grids (10,000 for THZ1 from two grids).

### Pre-processing of high-resolution datasets in cryoSPARC

**General pre-processing strategy.** Datasets were acquired and processed in cryoSPARC live (cryoSPARC versions 3.3.1-4.1.1)[34] with the same settings as for the screening datasets (see above). In addition to blob picking during on-the-fly processing, two parallel template-based picking strategies were then carried out after the end of the data collection, using representative 2D classes from the corresponding live-processing session or 2D projections generated from a 3D

reconstruction as templates (Supplementary Fig. 14). Template-based picks were separately extracted from the micrographs marked as accepted during live processing, using an extraction box size of 160 × 160 pixels, and classified into 200 2D classes (100-140 iterations, circular mask diameter 104–110 Å, batch size per class 200). For most datasets, the best 2D classes from the live processing session (blob picking) and from the two post-acquisition template-based picking strategies were selected, combined and duplicate particles were removed. The combined particles were re-extracted to improve particle centering and any duplicate particles arising from the re-centering during re-extraction were removed again. This final particle set was exported and converted to a *.star file compatible with RELION using a conversion program available in the PYEM package[44] followed by adjustment of path names to adhere to RELION conventions. This strategy was applied to all datasets with occasional small variations, outlined in the paragraphs below.

**CAK-THZ1**. For each of the two CAK-THZ1 datasets, which were collected early and used for trialing of processing strategies, the particles selected from the three picking strategies (1× blob picking, 2× template picking) were separately re-extracted before combination into one dataset, duplicate particle removal, and export to RELION format (combined particles). Additionally, the particles selected from blob picking and on-the-fly 2D classification in cryoSPARC live were exported as well (blob particles). The combined particles and the blob particles were separately imported into RELION, giving a total of four imported particle sets (two from each CAK-THZ1 grid).

**Apo-CAK**. For the apo-CAK complex, particles were additionally classified into 50 2D classes after the first duplicates removal step to remove any classes with concentric ring patterns arising from sensor imperfections from the selected particle set.

**CAK-ICEC0510-S**. The final duplicate particle removal step occurred within RELION after a 3D refinement.

### High-resolution data processing in RELION
**General refinement strategy**. All datasets were further processed using RELION 4.0 (Supplementary Fig. 14)[37]. To ensure full RELION functionality, including Bayesian polishing[35], movies were motion corrected using the RELION CPU-based implementation of the motion correction algorithm[45], using a binning factor of 2.0 (giving a pixel size of 1.14 Å/pixel). Exported particles from cryoSPARC were re-extracted from the motion corrected micrographs using an extraction box size of 160 × 160 pixels. Extracted particles were subjected to masked 3D refinement using a previous CAK reconstruction as an initial reference[15], with an initial angular sampling of 7.5 degrees. For ligand-bound datasets, the refined particles were classified into four 3D classes by alignment-free 3D classification ($\tau = 24$, mask diameter 104 Å, resolution E-step limited to 4 Å). The best class(es) were selected, refined, and subjected to Bayesian polishing[35], after first training the polishing on 10,000 particles, with re-windowing at a box size of 384 pixels and re-scaling to 256 pixels at 0.855 Å/pixel to account for CTF delocalization. Polished particles were refined and subjected to CTF refinement of per-particle defocus followed by refinement of beam tilt, trefoil, and fourth-order aberrations or vice versa (depending on which strategy provided the best result for each dataset)[36], followed by 3D refinement and a second round of polishing with re-windowing at a box size of 480 pixels and re-scaling to 384 pixels at 0.7125 Å/pixel. Re-polished particles were re-refined and the resulting map was post-processed in RELION. Deviations from this general strategy are outlined in the following paragraphs.

**CAK-ATPγS**. After completion of the processing pipeline outlined above, a final 3D classification step using 2 classes ($\tau = 36$) was inserted to identify the best-quality particle set used for the final refinement.

**CAK-LDC4297**. Due to sample heterogeneity, preferred orientation, and the presence of non-particle images, two additional 2D classification steps were inserted between the first 3D classification and the first Bayesian polishing step, selecting approximately 61,000 rare particle views to be added to the best 3D class, and eliminating non-particles from the resulting dataset. Subsequent polishing, refinements, and CTF refinements were performed according to the scheme outlined above. As a final step, a local refinement using a mask encompassing only CDK7 was performed to improve the appearance of the density of the inhibitor (Supplementary Fig. 2d, e).

**CAK-BS-181, CAK-ICEC0942 (grid VC13-3), CAK-ICEC0943**. The training step of Bayesian polishing failed, and particle polishing was instead run using the following ad hoc parameters, chosen based on the results of successful training runs from other samples, which used the same microscope and identical grid type: $\sigma_{vel} = 1.2$, $\sigma_{div} = 4500$, and $\sigma_{acc} = 1.5$.

**Apo-CAK**. Training for Bayesian polishing failed, and polishing was run with ad hoc parameters (as described above). Due to conformational heterogeneity of the β-sheet in the N-terminal kinase lobe in apo-CAK, particles were 3D classified while masking out the β-sheet after the initial motion correction, particle extraction, and initial refinement steps. This was done to focus the classification on the bulk of the particle density to select for particle quality, rather than letting the mobile β-sheet density dominate the classification. The selected particles were then refined using a full mask and polished only once (with re-windowing and re-scaling to a box size of 384 pixels at 0.855 Å/pixel) before a final refinement.

**CAK-THZ1**. Each of the two datasets (grids BG29-1, BG29-2) was motion corrected and each of the four exported particle sets from cryoSPARC (see above) were separately extracted from their corresponding micrographs. Extracted particles were separately refined and 3D classified, before selected particles were combined, duplicates were removed, and the resulting particle set refined before the initial polishing step. Training for Bayesian polishing failed, and polishing was run with ad hoc parameters (as described above). After the second polishing step, refined particles were additionally classified into 96 2D classes ($\tau = 2$, mask diameter 110 Å, E-step resolution limited to 6 Å) to remove non-particle classes before the final refinement.

**CAK-ICEC0510-S, CAK-dinaciclib**. After the first 3D classification step, particles were selected from two different iterations (best class from iteration 15, two best classes from iteration 25), refined separately, joined, and subjected to duplicate removal. The resulting particle set was refined and polished (using ad hoc parameters because training failed, see above). Polishing and re-refinement steps were carried out according to the standard procedure above, but a 3D classification (3 classes) was inserted between CTF refinement and the second polishing run to improve the quality of the final particle set.

**CAK-ICEC0574**. CTF refinement consisted of aberrations followed by per-micrograph astigmatism.

**CAK-ICEC0768**. CTF refinement included only aberrations (beam tilt, trefoil, 4th order aberrations) and an additional 3D classification (3 classes, $\tau = 24$) was performed after the two polishing runs to identify the final particle set.

**CAK-ICEC0880, CAK-ICEC0829**. Additional 10-class 3D classifications (without resolution E-step limit) were carried out following the second round of polishing for ICEC0880 and ICEC0829. For ICEC0880, this revealed two distinct inhibitor conformations; classes corresponding to these two conformations were selected and refined. For ICEC0829, classes with the clearest inhibitor density were selected, combined, and refined.

**CAK-ICEC0942 (grid VC14-1)**. An additional 3D classification step (4 classes, $\tau = 24$) after the first round of polishing and three rounds of CTF refinement (initially only refining beam tilt and third order aberrations, then defocus, then beam tilt and all aberrations) were performed.

**CAK energy filter comparison**. Only the first polishing step was performed for this dataset.

## Model building and refinement

The CAK-THZ1 model was generated by rigid-body fitting of the previously published model[15] (PDB ID 6XD3) into the post-processed map in UCSF ChimeraX[46]. This was followed by several rounds of manual rebuilding in COOT (version 0.9.6)[47], during which water molecules were manually modeled based on inspection of the density and hydrogen bond distances, and real-space refinement in PHENIX (versions 1.20, 1.21)[48]. To improve solvent modeling, we attempted to apply Segmentation-guided Water and Ion Modelling (SWIM)[49] to our model building procedure. This routine, as implemented in UCSF Chimera, added a large number of non-water atoms, such as $Na^+$ ions, to our models. Considering that even very high resolution X-ray crystal structures of other CDKs (e.g. CDK2 at 1 Å resolution, PDB ID 6Q4G[50]) crystallized in sodium-containing buffer do not contain modeled $Na^+$ ions, we decided to adhere to a more conservative approach and modeled all water-like densities as waters, with the exception of the unambiguously identifiable hydrated $Mg^{2+}$ ion coordinating ATPγS.

The refined CAK-THZ1 model was used as a starting point for building the remaining models, for which ligands and restraint files were generated using PHENIX eLBOW[51]. Ligands were manually modeled into the densities in COOT and models were adjusted and refined in the same way as for CAK-THZ1. For CAK-LDC4297, CAK-ICEC0942, and CAK-CT7030, the structure was additionally refined using the PHENIX-OPLS4 pipeline[52] using the SCHRODINGER 2021-4 software package, followed by re-refinement in PHENIX while using reference restraints on the ligand, which resulted in improved overall MOL-PROBITY score and model-to-map fit. The structure modeled into the 1.7 Å-resolution map of the combined dataset (Supplementary Fig. 6a, b) was refined using SERVALCAT (version 0.2.122)[53]. For several models, the structures were refined using reference restraints on individual residues, typically in less-well resolved regions, resulting in improved geometry and model-to-map fit of these residues or nearby water molecules. These were: cyclin H E141 and CDK7 Y169 of CAK-BS-181; CDK7 Y169 of CAK-BS-194; CDK7 D92 of CAK-ICEC0574; CDK7 E62 and Y169 of CAK-ICEC0829; CDK7 E68 of CAK-ICEC0880 (ring-up conformer); cyclin H L248 and CDK7 Y169 of CAK-ICEC0942; and cyclin H M237 of CAK-CT7030. Water molecules with Q-score <0.70 were excluded from the models, except in the case of the water that is hydrogen bonded to N06 of the ligand in the CAK-LDC4297 structure, which has a Q-score of 0.66 but whose presence is clearly supported by the density and the presence of equivalent water molecules in several other structures. Refinement statistics obtained from the comprehensive validation tool in PHENIX are provided in Supplementary Tables 5–11.

## Visualization of molecular models

Molecular models and cryo-EM maps were visualized in UCSF ChimeraX (versions 1.2.5)[46] and PyMOL (The PyMOL Molecular Graphics System, versions 2.5.2-2.5.5 Schrödinger, LLC) for analysis, interpretation, and preparation of figures.

## Further validation data

Local resolution maps, particle orientation distributions, and a second view of the ligands for all high-resolution CAK cryo-EM maps included in this manuscript are shown in Supplementary Figs. 15–17. Sphericity values to quantitatively assess the effects of preferred orientation on the cryo-EM reconstructions are provided in Supplementary Tables 5–11 and were computed using the 3D FSC validation server[54].

## Reporting summary

Further information on research design is available in the Nature Portfolio Reporting Summary linked to this article.

## Data availability

Cryo-EM maps generated from 1-hour Glacios screening in this study have been deposited to the Electron Microscopy Data Bank (EMDB) with accession codes EMD-17470, EMD-17471, EMD-17472, EMD-17473, EMD-17474, EMD-17475, EMD-17476, EMD-17477, EMD-17478, EMD-17479, EMD-17480, EMD-17481, EMD-17482, EMD-17483, EMD-17484, EMD-17485, EMD-17486, EMD-17487, EMD-17488, EMD-17489, EMD-17490, EMD-17491, EMD-17492, EMD-17493, EMD-17494, and EMD-17495. Cryo-EM maps resulting from 4-h Glacios screening have been deposited to the EMDB with accession codes EMD-17496, EMD-17497, EMD-17498, EMD-17499, EMD-17500, EMD-17501, EMD-17502, EMD-17503, EMD-17504, EMD-17505, EMD-17506, and EMD-17507. High-resolution cryo-EM maps generated in this study have been deposited to the EMDB using accession codes EMD-17129, EMD-17508, EMD-17509, EMD-17510, EMD-17511, EMD-17512, EMD-17513, EMD-17514, EMD-17515, EMD-17516, EMD-17517, EMD-17518, EMD-17519, EMD-17520, EMD-17521, EMD-17522, EMD-17523, EMD-17536, and EMD-17754. Atomic coordinates of high-resolution ligand-bound complexes generated in this study have been deposited to the Protein Data Bank (PDB) using identifiers 8ORM, 8P6V, 8P6W, 8P6X, 8P6Y, 8P6Z, 8P70, 8P71, 8P72, 8P73, 8P74, 8P75, 8P76, 8P77, 8P78, 8P79, 8P7L, and 8PLZ (as detailed in Supplementary Tables 5-11). Electron micrograph movies for selected datasets have been deposited to the Electron Microscopy Public Image Archive (EMPIAR) with accession codes EMPIAR-11793, EMPIAR-11799, EMPIAR-11800, EMPIAR-11807, EMPIAR-11821, and EMPIAR-11823. Atomic coordinate models used in this study are publicly available from the PDB under accession codes 1JKW, 3NS9, 4KD1, 5JQ5, 5JQ8, 6ATH, 6Q4G, 6XD3, 6Z4X. Source data are provided with this paper.

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

## Acknowledgements

We thank Fabienne Beuron for help with electron microscopy, Chris Richardson for help with high-performance computing, Gary Newton for providing the OPLS4 computing environment, and Claudio Alfieri for providing access to his GPU servers. We thank Lauren Knightley, Meirion Richards, Beth Jago, Maggie Liu, Amin Mirza, and the ICR Structural Chemistry team for compound analysis. B.J.G. was supported by a career development fellowship from the Medical Research Council of the UK (grant number MR/V009354/1) and V.I.C. was funded by an ICR PhD studentship. S.A. received funding by Cancer Research UK (grant numbers C37/A9335, C37/A12011, C37/A18784). K.J. was supported by a PhD studentship from OneMoreCity and Imperial College London.

## Author contributions

B.J.G, A.K. and S.A. designed the cryo-EM-based project. A.G.M.B and A.K.B. directed the inhibitor discovery program at Imperial College and Carrick Therapeutics, respectively. A.B., S.H.B.K, M.B. and B.S. synthesized CAK inhibitors and provided characterization data. K.J. characterized CAK inhibitors. B.J.G., V.I.C. and J.F. prepared the cryo-EM specimens and performed initial screening. A.F.K. and A.K. collected all cryo-EM data contained in the manuscript. V.I.C. and B.J.G. processed the cryo-EM data. V.I.C. interpreted the cryo-EM data under the guidance of B.J.G. B.J.G. and V.I.C drafted the first version of the manuscript, and all authors contributed to its final form.

## Competing interests

A.K. and A.F.K are employees of Thermo Fisher Scientific, the manufacturer of the electron microscopes used in this study. S.A., A.G.M.B, and A.B. are named inventors on patents concerning CDK7 inhibitors, which have been licensed to Carrick Therapeutics (U.S. patents US 9932344 B2, US 8067424 B2, US11857552 B2, US8507673 B2). They have received milestone payments from the patent licensing, and own shares in Carrick Therapeutics. S.A. has received research funding and reagents from Carrick Therapeutics. S.A. has also received research funding from Astra Zeneca. S.H.B.K. and B.S. are named inventors on patents concerning CDK7 inhibitors, which have been licensed to Carrick Therapeutics (U.S. patents US 9932344 B2, US 8067424 B2, US11857552 B2, US8507673 B2). They have received milestone payments from the patent licensing. M.B. has received milestone payments from the patent licensing. A.K.B. is an employee and shareholder of Carrick Therapeutics. All other authors declare no competing interests.
