## [Peer Review File · Nature Communications]

Reviewers' Comments:

Reviewer #1:

Remarks to the Author:

The manuscript by Cushing et al. describes a series of high-resolution cryo-EM structures of the CDK7-cyclin H-MAT1 complex (CDK-activating kinase, CAK) in its free and nucleotide-bound states and in complex with 15 different ATP-site directed (type I) small molecule inhibitors. A detailed structure-activity relationship (SAR) study of pyrazolopyrimidine-containing inhibitors revealed the structural basis of inhibitor selectivity for CDK7 over other CDKs, particularly CDK2 which is a known off-target of many CDK7 inhibitors. Previous efforts towards CDK7 inhibitor development were hampered due to the lack of robust crystallization conditions and cocrystal structure determination of CAK. Only a single crystal structure of CDK7 has been published in the PDB, but this was done in the absence of cyclin H and MAT1. The authors introduce an efficient workflow and convincingly demonstrate that current cryo-EM technology is suitable for routine 2 Å structure determination of nominally difficult targets in high-throughput. The manuscript is well written, the study design, methods and data analysis are robust, and the conclusions drawn are sound.

Minor points:

- Page 5, line 10: "In contrast to THZ1, which acts via a covalent mechanism, LDC4297 is a highly selective competitive inhibitor of CDK7." Albeit THZ1 is a covalent inhibitor, it is also ATP competitive. In this context it is better to contrast the different modes of action by using reversible vs. covalent (or irreversible). E.g., "In contrast to THZ1, which acts via a covalent mechanism, LDC4297 is a highly selective reversible inhibitor of CDK7."
- Fig. 4: The results of 1-hour Glacios screening is better suited for the supplementary information, as it is referred to only once in the text.
- Caption to Fig. 5, last sentence. Dinaciclib is not selective for CDK2; it is a rather promiscuous CDK inhibitor (CDK1, 2, 5 and 9).
- Fig. 6 e, f, g: ICEC1038 should be called dinaciclib as in Fig. 1 and throughout the text.
- Data availability section: The EMPIAR accession codes should be specified (currently XXXX)

Reviewer #2:

Remarks to the Author:

Cushing *et al.* present a number of electron cryomicroscopy (cryo-EM) structures of the 85 kDa CAK complex, with bound inhibitors. The authors have improved the resolution of the generated maps compared to their prior study, progressing from 2.5 Å (PMID: 33476598) to under 2.0 Å. Maps at this resolution reveal densities of water-molecule networks, visualization of which is a requirement for structure-based drug discovery. This paper presents a compelling exercise that showcases the power of the latest improvements in cryo-EM technology to enable drug discovery efforts. This manuscript will be of broad interest — both to the CDK community and generally to the structural biology / drug hunting audience interested in using cryo-EM in their work.

Some interesting features of this manuscript:

- Revealed that the nucleotide binding to human CAK was in a different conformation compared to what was reported for *Chaetomium thermophilum* (PDB ID: 6Z4X).
- Revealed that the elusive apo structure of CAK is in fact very similar to that of inhibitor-bound structures — with the exception of the N-terminal beta-sheet domain.
- Established benchmarks for cryo-EM screening and data collection of small-molecule-bound proteins on 200 kV vs 300 kV microscopes.
- Improved the structure of CAK-ICEC0942 — over the group's previous attempts.
- Shed some light on the molecular basis for inhibitor selectivity between CDK2 and CDK7 — by comparing the conformations of inhibitors bound to these kinases, and the small differences in

binding pockets.

Overall, this is a well-executed cryo-EM paper that established important benchmarks that were needed in the field to appreciate how much data is needed to begin using cryo-EM for drug discovery projects. The paper does very well on that front — technical comparisons between different collection strategies / microscopes are made, providing informative plots, etc. What is less impressive is the lack of quantitative comparison of reconstituted ligand densities — across the various maps that were presented in this project.

This reviewer recommends publication of this study. No additional experiments are recommended. However, a few instances of data presentation, analysis, and discussion, require extra work. Please see comments below.

MAJOR COMMENTS:

1. Statistics show that at similar resolving power, cryo-EM maps reveal fewer water molecules than maps from crystallography (PMID: 33087927 — see Extended Data Fig. 6b). Given how important accurate determination of water molecule positions is for drug discovery, the authors should discuss this in light of their own work, and perhaps provide a comparison to crystal structures of relevant complexes. While this reviewer appreciates that most deposited crystal structures of CDK7 are approx. 2.5 — 3.0 Å in resolution, this might not be a fair comparison. But perhaps for the purpose of this exercise the authors can use crystal structures of homologous CDKs that do reach <2.0 Å resolution. When “calling” water densities, the authors might consider using the criteria established by the Chiu lab (PMID: 33139928).

2. The authors report a number of maps of bound molecules to CAK. However, in some cases, the resolution of these molecules is not uniform across all atoms — for example in the case of THZ1. The authors conclude that this non-uniformity is likely due to flexibility of the bound molecule, and that extra data or further processing are unlikely to help resolve the weaker parts. Given the fact that the final reported particle sets are rather large (>0.5M), it would be useful if the authors expanded on this conclusion by showing how these extra things cannot help. The main narrative of this manuscript is that cryo-EM is now ready to enable drug discovery. This is one situation where authors can do better in explaining the limitations of the method in more detail.

3. Further discussion is needed on the differences in nucleotide engagement between human and *C. thermophilum* CAK. Is this likely a real difference or perhaps the 6Z4X model has an error? Have both modes been previously reported in high-resolution structures of kinases? And if so, is there a precedent that a single kinase (from two distant homologs) can engage nucleotides in two different modes?

4. There is no discussion about the partial rigidification of the N-terminal beta-sheet domain of CAK upon ligand binding. Is this expected / unique? Do other kinases also show this? What does this mean? Any functional consequences on substrate recruitment, etc?

5. Fig. 4 and Fig. 5 are not very informative and bring very little to the table in their current state. Both are collections of maps (plus FSC plots), with highlighted ligand densities. The authors should think of an additional more quantitative way to compare the quality of these various datasets — in a single plot. Something that does not require the reader to visually inspect every structure one-by-one. For example — Q scores for ligand fitting. Also, it would be useful to see a quantitative score plotted that illustrates the progression of how the ligand densities improve with more data.

6. Likewise, Fig 6. Is also rather abstract with very little information content. More thorough labeling should be provided, highlighting differences between compounds, and distance displacement. In order for this figure to truly accompany text, better renderings will be needed to show how CDK7 vs CDK2 selectivity works based on the differences in (i) binding pockets, (ii) compound conformations.

7. Instead of actually writing out all methods, on some occasions the authors take a shortcut and reference past papers — “*as described previously*”. This is bad practice. References can be kept, but full description of methods is expected of a serious research paper. This is important to ensure reproducibility, and also to avoid daisy-chaining of references, requiring readers to purchase multiple papers to get information that should be included in the original publication. Some examples of methods that this reviewer noticed require more attention: (i) compound synthesis, (ii) CAK expression, purification, (iii) covalent CAK-THZ1 sample preparation.

MINOR COMMENTS:

8. There is no need to use war-related terminology in chemistry. Please consider changing “warhead” to “reactive group”.

9. THZ1 is a covalent inhibitor. Perhaps consider showing this covalent engagement in Supp. Fig. 1l.

10. All structural figures need labels — of individual residues or secondary structure elements. There are many figures that have either zero or very little labeling provided: (i) Supp. Fig. 1n, 1m. (ii) Fig. 3b, 3c. (iii) Fig. 7a. (iv) Fig. 8b.

11. Supp. Fig. 1l needs a visual scale bar of Q scores.

12. Many figures are rasterized, which means that small labels are rather difficult to read — for example in Figs. 4, 5, 6. Please use text in the vector format if supported by the journal.

Reviewer #3:

Remarks to the Author:

The paper presents an impressive and substantial body of work resulting from the cryo-EM workflow the authors have established to determine multiple protein-small molecule inhibitor complex structures to support drug discovery. It will be of great interest to structural biologists, medicinal chemists and those interested in structure-aided inhibitor design. The significance of this work is two-fold:

- (i) High resolution structure determination of small complexes by cryo-EM at a throughput to support inhibitor design is feasible, and
- (ii) cryo-EM can support inhibitor design targeting small complexes.

The first point is well made: figures, tables and descriptions of the methods are included that support their workflow.

The paper would be improved if the second point was discussed and illustrated in more depth. The authors are in position to demonstrate that cryo-EM can generate structures of small complexes of comparable quality to those traditionally only thought to be accessible by X-ray crystallography and can answer the questions that are asked of structures for inhibitor design. For example, the paper includes structures that resolve enantiomers, and that analyse alternative binding modes and inhibitor conformations. Detailed enumeration of protein-inhibitor interactions is possible. To better make this point detailed analysis of the structures of both the bound inhibitor and the surrounding amino acids, waters and cofactors is required with the supporting experimental electron density maps for both protein and inhibitor structures included. For example, map quality is illustrated in Figure 1b-d, similar figures should be included in support of inhibitor binding mode. This level of detail is not required for all structures, it is suggested that a small number of exemplars are chosen and illustrated in the main text, with details for other inhibitors included in the supplementary information. Suggested figures and text that might be edited to address point (ii) are described in the list of minor points below.

Minor points.

1. p2, line 2: Ref 1 from 2005, consider adding an updated review by Fisher in 2019, PMID: 30488763.

2. p2, line 18: comparison where possible with crystal structures- evidence of cryo-EM providing access to populations unsampled from the X-ray analysis. Consider making the point that kinases are characterised by significant structural change upon cyclin binding, phosphorylation and substrate binding for catalysis. Cryo-EM is in a position to provide insights into these changes from in solution samples. Not just about sequence differences mediating specificity.
3. p3, line 13: For the 18 structures clarify how many different pharmacophore families- ie elaborate their structural diversity.
4. p4, line 12: Consider including a "resolution map" figure to accompany Figure 1a to show the local resolution. It would also be helpful to orient the view for non-kinase specialists- cyclin and CDK N and C-terminal lobes, MAT1 N- and C-termini all labelled. Consider adding this panel and removing either panel b or c.
5. Review all figures to ensure in each relevant amino acids are identified. For example, the other residues for which side chains are drawn in addition to W197 in Figure 1b, identify the acetylated residue and surrounding residues in Figure 1d; all residues surrounding the ATPyS in Figure S1f.
6. p4, line 16: Define regulatory T-loop (give residue range).
7. p4: line 26: Please define β -sheet domain. Residues and composition. Normally referred to as a β -sheet. Should the G-loop and the loop preceding the C-helix (as part of the N-terminal kinase lobe) be distinguished?
8. p4, line 23: Difference in adenine orientation in human vs *C. thermophilum* active sites. All kinases share a catalytic mechanism in which substrate binding and catalytically significant active site residues adopt a structure compatible with catalysis. That the two structures are different doesn't highlight the importance of using the human complex, likely that an in-solution method has selected an alternative (and possibly more functionally relevant) conformation. It would be interesting to compare both structures to that of a catalytically poised kinase complex.
9. p4, line 28, G-rich loop the loop that links β 1- β 2, define and give residue range to assist non-specialist readers.
10. p5, line 1. It would be helpful to identify the target amino acid sidechain covalently bound by THZ1 in the text. The warhead is clipped from the bottom of Figure S1l. Include all of the THZ1 structure and the target sidechain in a revised figure.
11. p5, line 13: Figure S1m: Label CDK7 and cyclin H.
12. p5, line 16: Figure S1n. Would it be possible to select orientations of panels Figure S1 l and n that would make this structural comparison easier to appreciate? Would zooming out from Figure S1l to include additional amino acids ease Figure S1l vs 1n comparison?
13. p5, line 25. Figure 2a, compounds in this figure all belong to one series, and so could be condensed into one exemplar, and then the standard R1, R2, R3 nomenclature for elaboration. Add R1, R2, R3 sub-structures into Table S2.
14. p6, line 6: Figure2 panels b-e. As panels c-e are representative maps from data collection for each of the steps in the work plan above, consider incorporating Figure 2 panels c-e into panel b, with arrows to link data collection details to maps?
15. Figures 4 and 5 summarise an impressive body of work, but the multiple panels are not structurally informative. I would suggest one Figure is included in the main text to support point (i) above and that the other two (? Figures 4 and 5) could be moved to the SI. To include representative structures in the main text, consider taking one representative compound and its associated panels from Figures 4 and 5 and add to Figure 3. To accommodate the extra Figure line, could panels (d and f), and (e and g) be combined?
16. p8, lines 15-20. Figure 6 would be improved by illustrating the interactions between the inhibitor and CDK7- for example enumerating the interactions and identifying the key amino acids. This elaboration might be easiest done by selecting a smaller number of key compounds and so fewer figure panels. It is difficult to discern the electron density that supports the inhibitor binding modes in Figure S6, though the associated Q-score rendering is informative. To address point (ii) above, would suggest Figure 6 revised to illustrate a smaller selection of inhibitors with panels that for each: (i) elaborate the molecular details of the inhibitor binding mode, (ii) include the electron density for the bound inhibitor and surrounding amino acids and (iii) provide the Q-score.
17. p8, line 22: Residues interacting with inhibitor should be identified in text and labelled in Figure 7a.
18. p21, line 24: Replace XXXXs with EMPIAR accession codes.

Cushing et al., Nature Communications, responses to reviewers

We thank all reviewers for their careful assessment of our manuscript and their detailed comments, which have guided us in improving the manuscript. Please find our detailed point-by-point answers below. We have highlighted all changes to the manuscript in yellow to facilitate review.

In addition to the changes suggested by the reviewers, we have re-acquired the cryo-EM dataset describing the CAK-CT7030 structure, which is now resolved at 1.9 Å (2.4 Å previously).

Reviewer #1 (Remarks to the Author):

The manuscript by Cushing et al. describes a series of high-resolution cryo-EM structures of the CDK7-cyclin H-MAT1 complex (CDK-activating kinase, CAK) in its free and nucleotide-bound states and in complex with 15 different ATP-site directed (type I) small molecule inhibitors. A detailed structure-activity relationship (SAR) study of pyrazolopyrimidine-containing inhibitors revealed the structural basis of inhibitor selectivity for CDK7 over other CDKs, particularly CDK2 which is a known off-target of many CDK7 inhibitors. Previous efforts towards CDK7 inhibitor development were hampered due to the lack of robust crystallization conditions and cocrystal structure determination of CAK. Only a single crystal structure of CDK7 has been published in the PDB, but this was done in the absence of cyclin H and MAT1. The authors introduce an efficient workflow and convincingly demonstrate that current cryo-EM technology is suitable for routine 2 Å structure determination of nominally difficult targets in high-throughput. The manuscript is well written, the study design, methods and data analysis are robust, and the conclusions drawn are sound.

We thank the reviewer for their positive assessment of our work.

Minor points:

- Page 5, line 10: “In contrast to THZ1, which acts via a covalent mechanism, LDC4297 is a highly selective competitive inhibitor of CDK7.” Albeit THZ1 is a covalent inhibitor, it is also ATP competitive. In this context it is better to contrast the different modes of action by using reversible vs. covalent (or irreversible). E.g., “In contrast to THZ1, which acts via a covalent mechanism, LDC4297 is a highly selective reversible inhibitor of CDK7.”

We agree with the reviewer. This change has been made as suggested.

- Fig. 4: The results of 1-hour Glacios screening is better suited for the supplementary information, as it is referred to only once in the text.

We have removed this figure. It is replaced by the corresponding panels in the supplementary information (with additional data on resolution and orientation distribution).

- Caption to Fig. 5, last sentence. Dinaciclib is not selective for CDK2; it is a rather promiscuous CDK inhibitor (CDK1, 2, 5 and 9).

We apologise for this oversight and have corrected the selectivity profile of dinaciclib.

- Fig. 6 e, f, g: ICEC1038 should be called dinaciclib as in Fig. 1 and throughout the text.

We thank the reviewer for pointing this out. This change has been made in what is now Figure 7.

- Data availability section: The EMPIAR accession codes should be specified (currently XXXX)

Due to the large volume of EMPIAR uploads (more than 30 TB), obtaining all accession codes took more time than anticipated. The codes have now been added to the manuscript:

“Electron micrograph movies for selected datasets have been deposited to the Electron Microscopy Public Image Archive (EMPIAR) with accession codes EMPIAR-11793, EMPIAR-11799, EMPIAR-11800, EMPIAR-11807, EMPIAR-11821, and EMPIAR-11823.”

Reviewer #2 (Remarks to the Author):

Cushing *et al.* present a number of electron cryomicroscopy (cryo-EM) structures of the 85 kDa CAK complex, with bound inhibitors. The authors have improved the resolution of the generated maps compared to their prior study, progressing from 2.5 Å (PMID: 33476598) to under 2.0 Å. Maps at this resolution reveal densities of water-molecule networks, visualization of which is a requirement for structure-based drug discovery. This paper presents a compelling exercise that showcases the power of the latest improvements in cryo-EM technology to enable drug discovery efforts. This manuscript will be of broad interest — both to the CDK community and generally to the structural biology / drug hunting audience interested in using cryo-EM in their work.

Some interesting features of this manuscript:

- Revealed that the nucleotide binding to human CAK was in a different conformation compared to what was reported for *Chaetomium thermophilum* (PDB ID: 6Z4X).
- Revealed that the elusive apo structure of CAK is in fact very similar to that of inhibitor-bound structures — with the exception of the N-terminal beta-sheet domain.
- Established benchmarks for cryo-EM screening and data collection of small-molecule-bound proteins on 200 kV vs 300 kV microscopes.
- Improved the structure of CAK-ICEC0942 — over the group’s previous attempts.
- Shed some light on the molecular basis for inhibitor selectivity between CDK2 and CDK7 — by comparing the conformations of inhibitors bound to these kinases, and the small differences in binding pockets.

Overall, this is a well-executed cryo-EM paper that established important benchmarks that were needed in the field to appreciate how much data is needed to begin using cryo-EM for drug discovery projects. The paper does very well on that front — technical comparisons between different collection strategies / microscopes are made, providing informative plots,

etc. What is less impressive is the lack of quantitative comparison of reconstituted ligand densities — across the various maps that were presented in this project.

This reviewer recommends publication of this study. No additional experiments are recommended. However, a few instances of data presentation, analysis, and discussion, require extra work. Please see comments below.

MAJOR COMMENTS:

1. Statistics show that at similar resolving power, cryo-EM maps reveal fewer water molecules than maps from crystallography (PMID: 33087927 — see Extended Data Fig. 6b). Given how important accurate determination of water molecule positions is for drug discovery, the authors should discuss this in light of their own work, and perhaps provide a comparison to crystal structures of relevant complexes. While this reviewer appreciates that most deposited crystal structures of CDK7 are approx. 2.5 — 3.0 Å in resolution, this might not be a fair comparison. But perhaps for the purpose of this exercise the authors can use crystal structures of homologous CDKs that do reach <2.0 Å resolution.

We found that even though they are homologous, the X-ray crystallographic structures of other CDKs differ in two important ways from our structure of the CAK: First, the surfaces accessible for water binding differ between structures due to the presence of different proteins, e.g. MAT1 for comparison to CDK-cyclin structures, or MAT1 and cyclin H for comparison of structures of the kinases in isolation. And second, the atomic-level details in these homologous but non-identical structures are different from those of CDK7, which leads to differences in water binding positions. The first point precludes straightforward comparisons of the number of modelled waters (such as the comparison referenced by the reviewer, though see below and Supplementary Fig. 2f, g in the manuscript), and the second point complicates water-by-water comparisons between structures. Nevertheless, when we compared the existing X-ray crystal structures of CDK7 and cyclin H to our CAK-THZ1 structure, we found that the identified water positions agree in many cases. We added the following comment to our manuscript:

“Knowledge of the positions of bound water molecules in macromolecular complexes is essential to understand inhibitor binding and to explore possibilities for inhibitor optimization¹⁹⁻²¹. Therefore, the visualization of water molecules is an important feature of our high-resolution cryo-EM maps. A systematic validation of our cryo-EM-based assignments by comparing them to the locations of water densities identified with X-ray crystal structures is challenging due to the lack of structures of CDK7 or cyclin H at resolutions of 2 Å or better. However, we find that many of the water locations identified in our cryo-EM maps correspond to those found in CDK7 and cyclin H at lower resolution or in related CDKs at high resolution (Supplementary Fig. 2f, g), supporting the reliability of our assignments. Nevertheless, when compared to X-ray crystallographic structures at similar resolutions, we observe that our cryo-EM structures appear to resolve a smaller total number of water molecules (approx. 160 as compared to nearly 400 in PDB ID 6ATH at 1.8 Å resolution²⁹). This has been observed previously for cryo-EM reconstructions at atomic resolution, but the reasons for this discrepancy are currently not known³⁰.”

When “calling” water densities, the authors might consider using the criteria established by the Chiu lab (PMID: 33139928).

We agree with the reviewer that water and ion modelling is a major challenge in the application of high-resolution cryo-EM to structure-based drug design, in part due to the inability of cryo-EM to unambiguously identify ions using anomalous scattering. Therefore, to improve our modelling efforts, we sought to apply the SWIM method described in the paper referenced by the reviewer to our maps and models. We found that this program, implemented in UCSF Chimera, added many sodium ions to our structures. Considering that even very high resolution X-ray crystal structures of other CDKs (e.g. CDK2: 6Q4G at 1 Å resolution) crystallised in sodium-containing buffer do not contain modelled sodium ions, we decided to adhere to a more conservative approach and modelled all water-like densities as waters, with the exception of the unambiguously identifiable hydrated Mg^{2+} ion coordinating $ATP\gamma S$. It is possible that our maps at roughly 2 Å resolution are not yet of sufficient quality for automated routines to function reliably, even though they do work effectively at higher resolutions (e.g. 1.3 Å for apoferritin). We foresee that our maps and models may aid methods developers in devising future algorithms for water and ion modelling in cryo-EM maps. We added the following statement to our methods section:

“To improve solvent modelling, we attempted to apply Segmentation-guided Water and Ion Modelling (SWIM)⁴⁷ to our model building procedure. This routine, as implemented in UCSF Chimera, added a large number of non-water atoms, such as Na^+ ions, to our models. Considering that even very high resolution X-ray crystal structures of other CDKs (e.g. CDK2 at 1 Å resolution, PDB ID: 6Q4G⁴⁸) crystallised in sodium-containing buffer do not contain modelled Na^+ ions, we decided to adhere to a more conservative approach and modelled all water-like densities as waters, with the exception of the unambiguously identifiable hydrated Mg^{2+} ion coordinating $ATP\gamma S$.”

2. The authors report a number of maps of bound molecules to CAK. However, in some cases, the resolution of these molecules is not uniform across all atoms — for example in the case of THZ1. The authors conclude that this non-uniformity is likely due to flexibility of the bound molecule, and that extra data or further processing are unlikely to help resolve the weaker parts. Given the fact that the final reported particle sets are rather large (>0.5M), it would be useful if the authors expanded on this conclusion by showing how these extra things cannot help. The main narrative of this manuscript is that cryo-EM is now ready to enable drug discovery. This is one situation where authors can do better in explaining the limitations of the method in more detail.

In our experience, it is not possible to classify for structural elements as small as a substituent on a small molecule with current cryo-EM data. While we cannot exclude that others may succeed in achieving this feat, we have never been able to do so. However, it is impossible to prove that this negative result always holds (or that our attempts were exhaustive), and we are therefore unable to satisfy this request of the reviewer. We have removed the statement in question and replaced it by referencing evidence supporting our interpretation regarding inhibitor flexibility:

“This is probably linked to continuous flexibility of this part of the inhibitor, a notion that is supported by the observation that the equivalent cysteine-reactive functional groups of two copies of the related inhibitor THZ531 are positioned 12 Å apart in the X-ray crystal structure of this compound bound to CDK12 ²⁷.”

3. Further discussion is needed on the differences in nucleotide engagement between human and *C. thermophilum* CAK. Is this likely a real difference or perhaps the 6Z4X model has an error? Have both modes been previously reported in high-resolution structures of kinases? And if so, is there a precedent that a single kinase (from two distant homologs) can engage nucleotides in two different modes?

We did verify that the deposited *syn*-conformation of the nucleotide is the conformation favoured by the X-ray crystallographic electron density of the *C. thermophilum* complex and concluded that a modelling error is unlikely. We have added further information and a more nuanced interpretation of these observations to the manuscript text:

“This conformation is rare, but has been observed previously in the structure of homoserine kinase ²³. It is currently unclear if the *syn*-conformation in the fungal complex is the preferred conformation in catalytically activated complexes, which could point to subtle differences in the nucleotide binding pockets and their small molecule-binding properties between the fungal and human enzyme, or if the *syn*-conformation in the *C. thermophilum* structure is favored only in the context of the protein conformation compatible with the crystal lattice.”

4. There is no discussion about the partial rigidification of the N-terminal beta-sheet domain of CAK upon ligand binding. Is this expected / unique? Do other kinases also show this? What does this mean? Any functional consequences on substrate recruitment, etc?

This is not unique and is expected, considering prior structural analysis of other kinases. We have added further information on this topic to the manuscript text:

“Ligand-dependent conformational changes of the N-terminal kinase lobe have been observed previously in cryo-EM structures of the human CAK ¹⁵, X-ray crystal structures of the homologous *C. thermophilum* complex ²⁴, and X-ray crystal structures of a range of other kinases, including CDK2 ²⁵ and cAMP-dependent protein kinase, where parts of this domain are displaced by almost 10 Å ²⁶. However, unlike X-ray crystal structures, where mobile domains can be trapped in specific conformations due to interactions within the crystal lattice and thereby manifest as defined conformations, cryo-EM structures can capture the full range of conformations accessible to molecular complexes in solution. This can explain the fragmented density observed in our cryo-EM map of apo-CAK.”

5. Fig. 4 and Fig. 5 are not very informative and bring very little to the table in their current state. Both are collections of maps (plus FSC plots), with highlighted ligand densities. The authors should think of an additional more quantitative way to compare the quality of these various datasets — in a single plot. Something that does not require the reader to visually inspect every structure one-by-one. For example — Q scores for ligand fitting. Also, it would be useful to see a quantitative score plotted that illustrates the progression of how the ligand densities improve with more data.

We appreciate this suggestion. We have removed Figure 4 and added a figure plotting the Q-scores of all structures as well as their ligands depending on workflow stage and resolution (now Supplementary Figure 5).

6. Likewise, Fig 6. Is also rather abstract with very little information content. More thorough labeling should be provided, highlighting differences between compounds, and distance displacement. In order for this figure to truly accompany text, better renderings will be needed to show how CDK7 vs CDK2 selectivity works based on the differences in (i) binding pockets, (ii) compound conformations.

We have added the new Figure 6 and new Supplementary Figure 11 and have extensively modified other Supplementary Figures (now Supplementary Figures 8-10) to show more detail of compound interactions and binding site differences.

7. Instead of actually writing out all methods, on some occasions the authors take a shortcut and reference past papers — “*as described previously*”. This is bad practice. References can be kept, but full description of methods is expected of a serious research paper. This is important to ensure reproducibility, and also to avoid daisy-chaining of references, requiring readers to purchase multiple papers to get information that should be included in the original publication. Some examples of methods that this reviewer noticed require more attention: (i) compound synthesis, (ii) CAK expression, purification, (iii) covalent CAK-THZ1 sample preparation.

We appreciate the reviewer’s point regarding completeness of methods. In some instances, the brevity of the methods section was chosen because certain experimental steps were not repeated during the course of this work. For example, for preparation of CAK-THZ1 grids, no new protein preparation was conducted. A CAK-THZ1 adduct protein sample prepared for the initial publication on CAK cryo-EM (Greber et al., PNAS, 2019) was retrieved from freezer storage and grids were prepared using this complex. We were reluctant to include detailed information for these experiments due to concerns that this would not accurately reflect the work conducted since this prior publication. However, as stated, we appreciate the equally important concern regarding completeness and reproducibility, and we have now added considerable amounts of information to the methods section. It would be impractical to paste all this information here due to its length, so we invite the reviewer to view the additional information in the manuscript (where it has been highlighted to facilitate review).

MINOR COMMENTS:

8. There is no need to use war-related terminology in chemistry. Please consider changing “warhead” to “reactive group”.

Even though the term “warhead” is standard terminology and used in several hundred papers indexed in PubMed, we appreciate the sentiment and have replaced the expression, now using: “the cysteine-reactive acrylamide group”.

9. THZ1 is a covalent inhibitor. Perhaps consider showing this covalent engagement in Supp. Fig. 1l.

This is now shown in Supplementary Figure 2a.

10. All structural figures need labels — of individual residues or secondary structure elements. There are many figures that have either zero or very little labeling provided: (i) Supp. Fig. 1n, 1m. (ii) Fig. 3b, 3c. (iii) Fig. 7a. (iv) Fig. 8b.

These figures have been extensively re-worked and contain more detailed labelling now.

11. Supp. Fig. 1l needs a visual scale bar of Q scores.

This has been added (now Supplementary Figure 2c). The same scale bar has been applied to Supplementary Figures 8-10 and 12.

12. Many figures are rasterized, which means that small labels are rather difficult to read — for example in Figs. 4, 5, 6. Please use text in the vector format if supported by the journal.

We have enlarged the labels, and the listed figures will be presented in high resolution in the final version of the manuscript (it is possible that they suffered during PDF compression during submission).

Reviewer #3 (Remarks to the Author):

The paper presents an impressive and substantial body of work resulting from the cryo-EM workflow the authors have established to determine multiple protein-small molecule inhibitor complex structures to support drug discovery. It will be of great interest to structural biologists, medicinal chemists and those interested in structure-aided inhibitor design. The significance of this work is two-fold:

- (i) High resolution structure determination of small complexes by cryo-EM at a throughput to support inhibitor design is feasible, and
- (ii) cryo-EM can support inhibitor design targeting small complexes.

The first point is well made: figures, tables and descriptions of the methods are included that support their workflow.

The paper would be improved if the second point was discussed and illustrated in more depth. The authors are in position to demonstrate that cryo-EM can generate structures of small complexes of comparable quality to those traditionally only thought to be accessible by X-ray crystallography and can answer the questions that are asked of structures for inhibitor design. For example, the paper includes structures that resolve enantiomers, and that analyse alternative binding modes and inhibitor conformations. Detailed enumeration of protein-inhibitor interactions is possible. To better make this point detailed analysis of the structures of both the bound inhibitor and the surrounding amino acids, waters and cofactors is required

with the supporting experimental electron density maps for both protein and inhibitor structures included. For example, map quality is illustrated in Figure 1b-d, similar figures should be included in support of inhibitor binding mode. This level of detail is not required for all structures, it is suggested that a small number of exemplars are chosen and illustrated in the main text, with details for other inhibitors included in the supplementary information. Suggested figures and text that might be edited to address point (ii) are described in the list of minor points below.

We thank the reviewer for their assessment of our paper, and we have incorporated the suggestions made in the detailed remarks into the manuscript.

We would like to note that in our view, the aspect of our work that provides the strongest support for the utility of cryo-EM in structure-based drug design (i.e. point (ii) in the reviewer's nomenclature) is the fact that we were able to (a) mechanistically explain why ICEC0942 is a better CDK7 binder than other, very similar inhibitors and (b) derive a mechanism contributing to inhibitor selectivity and correctly predict that addition of bulky groups to the benzylamine ring would decrease CDK2 inhibition.

Minor points.

1. p2, line 2: Ref 1 from 2005, consider adding an updated review by Fisher in 2019, PMID: 30488763.

This reference has been included, now reference #2 in the manuscript.

2. p2, line 18: comparison where possible with crystal structures- evidence of cryo-EM providing access to populations unsampled from the X-ray analysis. Consider making the point that kinases are characterised by significant structural change upon cyclin binding, phosphorylation and substrate binding for catalysis. Cryo-EM is in a position to provide insights into these changes from in solution samples. Not just about sequence differences mediating specificity.

We thank the reviewer for this suggestion. We have added one statement to the first paragraph of the introduction:

“Furthermore, CDKs are characterized by large conformational changes upon their association with their activatory cyclins and phosphorylation of their T-loops, which can change their compound-binding properties¹³.”

And we then refer to this in the second paragraph:

“Furthermore, one advantage of cryo-EM is the ability to capture a series of different states from a single sample, which can facilitate the structural analysis of dynamic systems. This may be particularly beneficial for CDKs due to their conformation-dependent compound-binding properties¹³.”

3. p3, line 13: For the 18 structures clarify how many different pharmacophore families- ie elaborate their structural diversity.

We have consulted with experts in drug design and computational chemistry at the ICR Centre for Cancer Drug Discovery and were advised that the term “pharmacophore families” is ambiguous and not well defined. Instead of using this terminology, we have added a statement listing the inhibitor families the compounds in our structures belong to:

“..., with the inhibitors comprising pyrazolopyrimidine, pyrazolotriazine, and phenylaminopyrimidine class compounds.”

4. p4, line 12: Consider including a “resolution map” figure to accompany Figure 1a to show the local resolution. It would also be helpful to orient the view for non-kinase specialists- cyclin and CDK N and C-terminal lobes, MAT1 N- and C-termini all labelled. Consider adding this panel and removing either panel b or c.

We have incorporated both suggestions, which can now be found in Figure 1a and c.

5. Review all figures to ensure in each relevant amino acids are identified. For example, the other residues for which side chains are drawn in addition to W197 in Figure 1b, identify the acetylated residue and surrounding residues in Figure 1d; all residues surrounding the ATP γ S in Figure S1f.

We have added more labelling throughout the manuscript, as also suggested by another reviewer.

6. p4, line 16: Define regulatory T-loop (give residue range).

We have added this information: “... density for the regulatory CDK7 T-loop (residues 155-182 between the conserved DFG and APE motifs) becomes fragmented...”

7. p4: line 26: Please define β -sheet domain. Residues and composition. Normally referred to as a β -sheet. Should the G-loop and the loop preceding the C-helix (as part of the N-terminal kinase lobe) be distinguished?

We have reworded this to “the β -sheet in the N-terminal kinase lobe”. The G-rich loop moves along with the remainder of the domain and is included in this designation.

8. p4, line 23: Difference in adenine orientation in human vs *C. thermophilum* active sites. All kinases share a catalytic mechanism in which substrate binding and catalytically significant active site residues adopt a structure compatible with catalysis. That the two structures are different doesn't highlight the importance of using the human complex, likely that an in-solution method has selected an alternative (and possibly more functionally relevant) conformation. It would be interesting to compare both structures to that of a catalytically poised kinase complex.

We acknowledge that the existing discussion of this observation was too cavalier and appreciate the opportunity to elaborate on this further. We have added the following statement to the text:

“This conformation is rare, but has been observed previously in the structure of homoserine kinase²³. It is currently unclear if the *syn*-conformation in the fungal complex is the preferred conformation in catalytically activated complexes, which could point to subtle differences in the nucleotide binding pockets and their small molecule-binding properties between the fungal and human enzyme, or if the *syn*-conformation in the *C. thermophilum* structure is favored in the context of the protein conformation compatible with the crystal lattice.”

9. p4, line 28, G-rich loop the loop that links β 1- β 2, define and give residue range to assist non-specialist readers.

We have now more accurately defined the terminology used:

“... outermost two β -strands, connected by the G-rich loop (residues 19-24, sequence GEGQFA), which are barely visible in the density.”

10. p5, line 1. It would be helpful to identify the target amino acid sidechain covalently bound by THZ1 in the text. The warhead is clipped from the bottom of Figure S1l. Include all of the THZ1 structure and the target sidechain in a revised figure.

Cysteine 312 is now mentioned in the text (“... the extended arm that harbors the reactive acrylamide group that covalently modifies CDK7 residue C312...”), and the clipping of the reactive group has been removed and it is now shown in Supplementary Figure 2a.

11. p5, line 13: Figure S1m: Label CDK7 and cyclin H.

These labels have been added (now Supplementary Figure 2d).

12. p5, line 16: Figure S1n. Would it be possible to select orientations of panels Figure S1 l and n that would make this structural comparison easier to appreciate? Would zooming out from Figure S1l to include additional amino acids ease Figure S1l vs 1n comparison?

This suggestion has been incorporated (now Supplementary Figure 2b, e).

13. p5, line 25. Figure 2a, compounds in this figure all belong to one series, and so could be condensed into one exemplar, and then the standard R1, R2, R3 nomenclature for elaboration. Add R1, R2, R3 sub-structures into Table S2.

We have added the suggested inhibitor core with R_x-substituent designations to our figure, and we have added the substituents to Supplementary Table 2. The suggested R_x-designation is used in the text and has facilitated discussion of certain compounds, demonstrating its utility. However, instead of fully relegating the information on compound structure to the supplementary materials as suggested, we prefer to keep the full inhibitors visible in the main

text as well (and considering that we are below the journal limits for the number of figures, we believe that this version of the figure can appropriately accommodate).

14. p6, line 6: Figure 2 panels b-e. As panels c-e are representative maps from data collection for each of the steps in the work plan above, consider incorporating Figure 2 panels c-e into panel b, with arrows to link data collection details to maps?

We have added the arrows to indicate the relationships between panel b and the subsequent panels, but we believe it is better to retain the panel labels c-e because it is awkward to refer to these panels from the text otherwise (e.g. for panel d, this would then have to be “panel b, bottom, middle” or similar, which is not practical).

15. Figures 4 and 5 summarise an impressive body of work, but the multiple panels are not structurally informative. I would suggest one Figure is included in the main text to support point (i) above and that the other two (? Figures 4 and 5) could be moved to the SI. To include representative structures in the main text, consider taking one representative compound and its associated panels from Figures 4 and 5 and add to Figure 3. To accommodate the extra Figure line, could panels (d and f), and (e and g) be combined?

We have moved Figure 4 to the supplementary information, as suggested. We note that the suggested compilation of cryo-EM maps for one inhibitor across data collection strategies was already shown in Figure 2 (and remains in place). The size of Figure 3 has been reduced because we combined the panel showing the resolution histogram with the new Q-score figure (Supplementary Figure 5).

16. p8, lines 15-20. Figure 6 would be improved by illustrating the interactions between the inhibitor and CDK7- for example enumerating the interactions and identifying the key amino acids. This elaboration might be easiest done by selecting a smaller number of key compounds and so fewer figure panels. It is difficult to discern the electron density that supports the inhibitor binding modes in Figure S6, though the associated Q-score rendering is informative. To address point (ii) above, would suggest Figure 6 revised to illustrate a smaller selection of inhibitors with panels that for each: (i) elaborate the molecular details of the inhibitor binding mode, (ii) include the electron density for the bound inhibitor and surrounding amino acids and (iii) provide the Q-score.

Most of the information suggested by the reviewer is already available in the manuscript, though we appreciate that it may have been difficult to inspect the Coulomb potential densities supporting our inhibitor assignments in what was formerly Supplementary Figure 6 at the previously submitted size. To facilitate their visualisation, we have enlarged the panels in Supplementary Figure 6, which has now been split into 3 separate supplementary figures that each fit on one page. We have also added panels that provide detailed information on inhibitor interactions, as suggested, to form the new main text Figure 6 and Supplementary Figure 11. These new figures are accompanied by additional text, which is highlighted in the revised manuscript (pages 10-11).

17. p8, line 22: Residues interacting with inhibitor should be identified in text and labelled in Figure 7a.

This panel is now part of Figure 6 and has been supplemented with more labelling. The discussion of inhibitor interactions has been expanded substantially (pages 10-11 of the revised manuscript).

18. p21, line 24: Replace XXXXs with EMPIAR accession codes.

Due to the large volume of EMPIAR uploads (on the order of 30 TB), obtaining all accession codes took some time. The codes have now been added to the manuscript:

“Electron micrograph movies for selected datasets have been deposited to the Electron Microscopy Public Image Archive (EMPIAR) with accession codes EMPIAR-11793, EMPIAR-11799, EMPIAR-11800, EMPIAR-11807, EMPIAR-11821, and EMPIAR-11823.”

Reviewers' Comments:

Reviewer #2:

Remarks to the Author:

The authors made a solid effort to address all comments and suggestions from this reviewer. The reviewer is satisfied by the responses and recommends this manuscript for publication.

Reviewer #3:

Remarks to the Author:

I have reviewed the responses to my initial review of this manuscript and can confirm that the authors have addressed them in their revised manuscript.